# Memory justifications provide valid indicators of retrieval accuracy across time
Avi Gamoran [1,2,5], Zohar Raz Groman [2,3,5] ✉, Michael Gilead [4] & Talya Sadeh [1,2,3]

Human beings share in others' experiences and learn from them, but epistemic vigilance is necessary to avoid being misled by false information, and to distinguish between veridical and non-veridical memories. Memory Justifications, individuals' explanations for why they believe a recalled event truly occurred, help maintain epistemic vigilance regarding our memories. Understanding how justifications are affected by the passage of time is crucial since they serve to ensure memory validity in everyday life and in legal settings. Using behavioral measures and linguistic analyses of participants' (N = 421) self-reported memory justifications, we examined changes in justifications' content and detail over time. The credibility of justifications was validated by comparing them with free recall performance. Results demonstrated a decrease in overall recall over time. However, the degree of episodic detail in justifications was steady across time delays, indicating preserved justification content over time. Pre-registered and exploratory analyses showed that the proportion of justified recalls and justifications' term frequencies were also preserved over time. Our findings suggest that individuals' memory justifications serve as relatively reliable indicators of retrieval accuracy, which remain stable over time. Still, lexical measures demonstrated that some aspects of justifications' content show subtle delay-related changes, which might be explained in terms of a time-dependent decline in subjective confidence.

Human beings have the highly adaptive ability to learn about things they did not experience directly[1,2]. This ability is believed to rely on the unique capacity of using language to guide human imagination[3,4]. However, the dependence on social learning and imagination also introduces challenges, as people may misremember events. To overcome such potential hazards, humans need to uphold constant "epistemic vigilance" to discern which information received from others is valid[5,6]. Furthermore, epistemic vigilance is required for justifying one's own recollections[7–10]. Vigilance regarding memory from our own past experience is in fact warranted, due to our capacity to imagine things which were not experienced directly[11–14]. How can we distinguish between veridical and non-veridical recollections? What evidence can we give to justify claims about recollected events?

Previous studies of retrieval of true memories (i.e., actual past events), and their distinction from false memories (i.e., events imagined or falsely recalled), demonstrate that both these types of memories share much in common, to the point where it has been suggested they are at

times indiscernible from one another[15–17]. However, in most cases, there are noticeable differences between true and false memories. Most importantly, true memories are associated with higher degrees of recollection. Recollection in this context refers to retrieval of past information along with vivid contextual information from the experience and often accompanied by high confidence. Previous studies have demonstrated subjective phenomenological differences between retrieval of true and false memories[18–20], as indicated by subjective ratings[16,19,21,22]. Retrieval of true memories is accompanied by details relating to sensory perception and embedding of an event in a spatial and temporal context[16–18,21,23–26]. The subjective differences between retrieval of true and false memories have further been demonstrated by objective measures of neural activity[17,21,27], perceptual priming[21] and linguistic analysis[28–31].

These findings suggest that recollection, the accompaniment of memories with contextual information, is an adaptation meant to address the fundamental problem of distinguishing between veridical and non-

[1]Department of Psychology, Ben-Gurion University of the Negev, Beer Sheva, Israel. [2]The School of Brain Sciences and Cognition, Ben-Gurion University of the Negev, Beer Sheva, Israel. [3]Department of Industrial Engineering & Management, Ben-Gurion University of the Negev, Beer Sheva, Israel. [4]The School of Psychology Sciences, Tel Aviv University, Tel-Aviv, Israel. [5]These authors contributed equally: Avi Gamoran, Zohar Raz Groman. ✉e-mail: zohargro@post.bgu.ac.il

veridical memories[8,32–34] (but see refs. 32–34). When we recall an event from memory, we do not merely recall its details, but often reinstate our sense of experiencing the event as well ("*autonoetic consciousness*"[34,35]). Information regarding one's internal context includes among other, our attitudes regarding the event (e.g., enjoyment from seeing a painting)[36] and our internal monologue (e.g., "I wonder who painted it?")[37]. The ability to reinstate an experience along with accompanying contextual information (internal or external) may serve as an authentication device that helps us determine that we are remembering correctly[8].

Thus, the contextual information present in memory accompanied by recollection is used as *justification* for the veracity of our memories. For example, when Mary remembers having gone to vote one blistery afternoon, she may ask herself how she knows she is properly recalling election day. In which case, she may provide some cues based on contextual information, such as the chill felt in the air, and remembering thinking to herself: "It's unusually chilly for this time in November", which will be brought forth as evidence for the memory's validity. Similarly, people may rely on others' recollection to evaluate the veracity of their memories[38]. For example, a juror may reason: "The witness is able to recollect his internal thoughts during the incident and retold the event in vivid detail as if he is reexperiencing it, so he is likely describing a true memory".

Indeed, recent research has demonstrated that memory justifications can be used for memory verification[39,40]. The natural language used in written justifications for decisions in a recognition task can be modeled to separate accurate memories from false ones on a recognition task[31,41]. Furthermore, eyewitnesses' justifications in a line-up identification can be used to classify true and false recognitions with reasonable certainty[42]. These findings demonstrate that there is valuable information retained in such justifications, as employed by the classification models. Importantly, this information is also used by humans as well in validating their own memories[31,43], and in corroborating others' memories[30,44].

As is true for most of the information stored in memory, it is most likely that the contextual cues on which justifications rely are subject to the perils of time[45–47]. Previous research has demonstrated the effects of time on rankings of confidence and metacognitive judgements[48–50]. However, despite the importance of verbal justifications in evaluation of memory's veracity, including in highly consequential settings[42], research examining how they are affected by the passage of time remains limited. Examination of the effect the passage of time has on justifications validity is crucial because retrieving information from memory is typically done after a delay. How then are justifications affected by time?

Intuitively, one may reason that contextual information upon which justifications rely, like other information stored in memory, fragment over time, losing more details and clarity as time goes by refs. 46,51,52. Furthermore, previous studies examining the effects of time on confidence, and metacognitive assessments of memory, have demonstrated that subjective confidence and the calibration between subjective confidence and objective accuracy weakens over time[53,54]. In contrast, recent studies have provided evidence for cases in which accessibility of memories is reduced over time, but the fidelity and clarity of an accessible memory remains intact[45,55–58]. In the current study, we ask whether justifications lose fidelity over time, in line with findings of subjective confidence assessments, or whether they remain intact in their degree of vividness, detail, and their content after a long time-delay. A *decrease* in fidelity, vividness and detail over time would be expected due to a graded decaying of underlying contextual information, whereas *maintained* levels of fidelity, vividness and detail would be in line with a dissociation of accessibility from fidelity. In the latter case, despite reduced accessibility to contextual information, that information which is still accessible will remain unchanged in degree of fidelity (demonstrating an "All-or-None" form of forgetting).

Conversely, an alternate explanation for an all-or-none pattern of results is that the content of some contextual information is prone to degradation over time, whereas other information remains stable over time. The contextual information which does not degrade would remain accessible and with maintained levels of fidelity. If an all-or-none pattern of forgetting is found, future studies will be needed to elucidate between these two theoretical accounts.

To examine the effects of time on memory justifications, we gathered justifications regarding items participants recall by explicitly asking them to detail why they think the particular item appeared in the study phase and to justify their recollection of it. A standard delayed free recall paradigm for word lists was employed. Following each recall test, individuals were asked to specify the subjective information that prompted their recall of each of the words.

To examine fidelity of justifications over time, the online recall test was conducted either after a short (1.5 min), or a long (one day) delay period (See Fig. 1). We conducted a validation check to examine the main study manipulation by testing whether information was forgotten over time. We predict a decrease in the number of items recalled over time in line with delay-dependent forgetting[27]. We conditioned further analysis of the study hypotheses on confirmation of this validation check.

Our analyses assessed three hypotheses (which have also been examined in the pilot data, see Supplementary Note 1 | Pilot study 1). These are described in Table 1 below. Our first hypothesis relates to the link between subjective justifications and recall performance, establishing that memory justifications indeed reflect recall performance. First, we predicted that

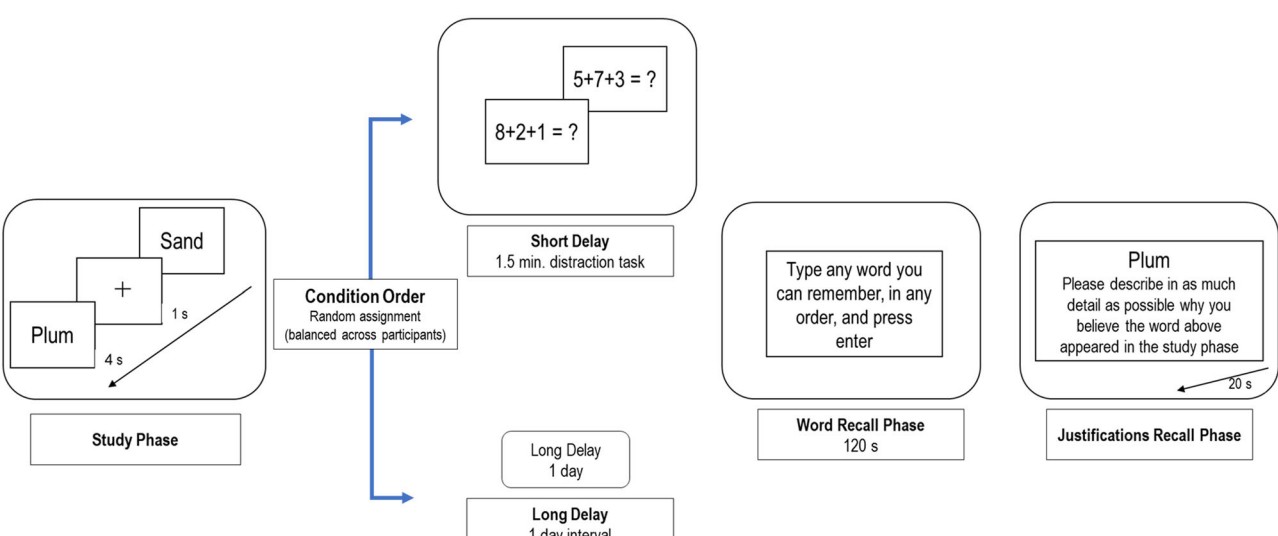

**Fig. 1 | Illustration of the experimental sequence.** Schematic depiction of the study and test phases across the Short- and Long-delay conditions.

**Table 1 | Hypothesis table**

| Question | Hypothesis | Sampling plan (e.g., power analysis) | Analysis Plan | Interpretation given to different outcomes |
|---|---|---|---|---|
| H1. Do justifications reflect mnemonic performance? | Higher recall accuracy will be found for items recalled with a justification than items without a justification, supporting the claim that justifications reflect mnemonic performance. | - | Items recalled per subject will be assessed as either correct/incorrect, and manually coded as including justifications/not including justifications. A GLMM model will be used for statistical inference of the hypothesis. | Model comparison will be conducted between GLMM models with and without the delay main effect, by comparing the differences in BIC scores and converting to a Bayes Factor score. A $BF_{10} > 10$ will be interpreted as supporting higher recall accuracy for items recalled with justifications than items recalled without justifications supporting the claim that justifications reflect mnemonic performance. A $BF_{10} < 0.1$ ($BF_{01} > 10$) will be interpreted as providing evidence of a null effect with no higher recall accuracy for items recalled with a justification. $0.1 < BF_{10} < 10$, will be interpreted as inconclusive results. |
| H2. Is the link between justifications and recall accuracy affected by study-test delay? | No significant interaction will be found with time delays, such that accuracy for items with justifications will not decrease over time, further supporting the validity of justifications. | - | Items recalled per subject will be assessed as either correct/incorrect, and manually coded as including justifications/not including justifications. A GLMM model including both main effects and their interaction will be used for statistical inference of the hypothesis. | Model comparison will be conducted between a GLMM model including both main effects as well as the interaction effect between delay and justifications, and a model including both main effects but without the interaction effect, by comparing the differences in BIC scores and converting to a Bayes Factor score. A $BF_{10} > 10$ will be interpreted as supporting the existence of an interaction effect on recall accuracy between time delays and the existence of justifications, supporting the claim that the link between justifications and recall accuracy is differentially affected according to time delay. A $BF_{10} < 0.1$ ($BF_{01} > 10$) will be interpreted as providing evidence of a null effect with no interaction effect between time delays and justifications. $0.1 < BF_{10} < 10$, will be interpreted as inconclusive results. |
| H3. Is justifications' content preserved over time? | Either a decline in justifications' level of detail, for correctly recalled justifications, from short to long delays, supporting a fragmenting in contextual information over time, or a similar level of detail will be found in the short and long delays, and will be taken as support for preserved contextual information over time. | Up to a Bayes Factor score of 10, either in support of the null, or in support of the difference hypothesis, or until reaching an N of 500 participants. | Items recalled per subject will be assessed as either correct/incorrect, and manually coded as including justifications ($J+$) or not including justifications ($J-$). Level of detail per justification will be quantified with the automated autobiographical interview score of internal and external details. Bayesian t-tests will be used for inference of the hypothesis | A $BF_{10} > 10$ will be interpreted as supporting a fragmenting in justifications over time, such that fewer details are recalled following the long delay than the short delay. A $BF_{10} < 0.1$ ($BF_{01} > 10$) will be interpreted as providing evidence of a null effect that justifications are recalled with the same detail for both delays. $0.1 < BF_{10} < 10$, will be interpreted as inconclusive results. |
| E1. Are proportions of justifications steady over time? | Either a decline in the proportion of items with justifications out of all items recalled will be found from short to long delays, supporting a fragmenting in contextual information over time, or a similar proportion of items with justifications will be found in the short and long delays, and will be taken as support for preserved contextual information over time. | - | Items recalled per subject will be assessed as either correct/incorrect, and manually coded as including justifications/not including justifications. Bayesian binomial-tests will be used for inference of the hypothesis | A $BF_{10} > 10$, will be interpreted as supporting a difference in proportion of items recalled with justifications between delay conditions. This would support a shift of justifications' content over time. A $BF_{10} < 0.1$ ($BF_{01} > 10$) will be interpreted as providing evidence of a null effect such that proportion of items recalled with justifications are similar for both delays. |

**Table 1 (continued) | Hypothesis table**

| Question | Hypothesis | Sampling plan (e.g., power analysis) | Analysis Plan | Interpretation given to different outcomes |
|---|---|---|---|---|
| E2. Are Justifications' contents, as reflected in lexical measures (concreteness, tentativeness, certainty), similar across time delays? | Either a decline in lexical measures of justifications' concreteness, and confidence, will be found from short to long delays, supporting a fragmenting in contextual information over time, or similar levels of justifications' concreteness, and confidence will be found in the short and long delays, and will be taken as support for preserved contextual information over time. | - | Justifications will be assessed using lexicons of language concreteness, certainty and tentativeness. Bayesian t-tests will be used for inference of the hypothesis | A $BF_{10} > 10$, will be interpreted as supporting a difference in levels of lexical measures of justifications between delay conditions. This would support a shift of justifications content over time. A $BF_{10} < 0.1$ ($BF_{01} > 10$) will be interpreted as providing evidence of a null effect such that lexical measures of justifications remain similar over time, in support of a preservations of justifications' content over time. $0.1 < BF_{10} < 10$, will be interpreted as inconclusive results. |
| E3. Are Justifications' contents, reflected in unigram frequencies, similar across time delays? | Either a shift in lexical measures of justifications' unigram frequencies, will be found from short to long delays, reflecting changes in justifications' content and supporting a fragmenting in contextual information over time, or similar levels of justifications' unigram frequencies will be found in the short and long delays, and will be taken as support for preserved contextual information over time. | - | Frequencies of unigrams used in justifications will be assessed, and all unigrams above a cutoff of a term frequency of 5 will be included (excluding rarely used terms). Bayesian t-tests will be used for inference of the hypothesis. | A $BF_{10} > 10$, will be interpreted as supporting a difference in levels of justifications' unigram frequencies between delay conditions. This would support a shift of justifications content over time. A $BF_{10} < 0.1$ ($BF_{01} > 10$) will be interpreted as providing evidence of a null effect such that unigram frequencies remain similar over time, in support of a preservation of justifications' content over time. $0.1 < BF_{10} < 10$, will be interpreted as inconclusive results. |

justifications will reflect objective memory performance, such that items with justifications will be more likely to be accurate recalls compared to items without Justifications (Hypothesis 1). With regard to effects of time, we further predicted that the accuracy of items with justifications will not decrease over time, contrary to what would be predicted if justifications of true memories are prone to increased confabulation over time, in which case the proportion of items with justifications which are correct recalls ought to decrease (Hypothesis 2).

The main study hypothesis concerns the question of whether time-dependent forgetting of contextual information, information which underlies memory justification, displays a graded course of forgetting or conversely an all-none pattern of forgetting[45,55,58–61]. The level of detail in justifications for correctly recalled items (i.e., justifications of true memories), as measured by the automated autobiographical interview[62–64] were used to arbitrate between these two differing accounts. Justification details after Long versus Short time delays will either change ($BF_{10} > 10$), in line with graded forgetting, or remain unchanged ($BF_{01} > 10$), supporting an all-or-none forgetting (Hypothesis 3).

We further set out an additional three exploratory analyses regarding the content of justifications over time. Along a similar vein to the shifting or preservation of justification content over time, we further hypothesized that the proportion of items reported with justifications may be informative in discerning between graded forgetting of contextual information or all-or-none forgetting. The proportion of items with justifications may differ ($BF_{10} > 10$) between short and long delays, supporting graded forgetting. Conversely, the proportions of items with justifications may remain similar ($BF_{01} > 10$) in short and long delays, supporting all-or-none forgetting (Exploratory analysis 1).

Finally, we assessed justifications' content using linguistic analysis. Specifically, we employed domain-specific lexicons to examine similarity in relevant topics, and frequencies of single words ("unigrams") for a measure of similarity in words used within justifications. We used lexicons relevant to justifications' degree of vividness and confidence, namely, language concreteness[65,66], and tentative versus certain language[67,68]. We hypothesized that measures of language concreteness and certainty will either decline from short to long delays ($BF_{10} > 10$) lending support to the degradation of underlying contextual information, or they may remain similar ($BF_{01} > 10$) in short and long delays, supporting an all-or-none forgetting and preserved contextual information over short and long time delays (Exploratory analysis 2).

We further hypothesized that unigram frequencies[69,70] will either shift from short to long delays ($BF_{10} > 10$), reflecting changes in justifications content, and supporting degradation of contextual information over time. Alternatively, unigram frequencies may remain similar ($BF_{01} > 10$), reflecting broadly similar justifications' contents between long and short time-delays, and supporting preservation of contextual information (Exploratory analysis 3)

## Methods
### Protocol registration
The Stage 1 protocol for this Registered Report was accepted in principle on January 23, 2025. The protocol, as accepted by the journal, can be found at: https://osf.io/s6qv8.

### Ethics information
The research complies with ethical regulations and was approved by the institutional review board at Ben-Gurion University of the Negev, Beer-Sheva, Israel. The participants provided written consent prior to the research. Participants were provided with monetary compensation according to a rate of £8 per hour.

### Design
**Participants.** 509 Participants (native English speakers aged 18–35) were recruited from Prolific (https://www.prolific.co/), a web-based participant pool for behavioral studies. Participants had normal or corrected-to-normal vision (by their reporting) and were compensated at a rate of £8 per hour. The study was approved by the ethics review board at Ben-Gurion University of the Negev, Beer-Sheva, Israel.

88 participants were excluded from the study due to misunderstanding task instructions. The remaining 421 were included in the analyses. All participants self-reported their gender (232 female, 189 male). Data on race or ethnicity were not collected.

### Materials
**Lists.** The experiment consisted of two lists of 16 words each. A fixed list of 16 words served as a practice-list, presented before the first block and not included in the analyses. Words are 3–10 letters long nouns and adjectives, selected from the Penn Electrophysiology of Encoding and Retrieval Study (PEERS)[71,72] word pool, which contains 1638 words (available at: http://memory.psych.upenn.edu/files/wordpools/PEERS_wordpool.zip). The lists have been constructed such that temporal and semantic contributions to recall can be dissociated. In each list, varying degrees of semantic relatedness occur at both adjacent and distant serial positions. Semantic relatedness was determined using the word association space ("WAS"[73]). The WAS similarity values were used to group words into four similarity bins (high: $\cos\theta > 0.7$; medium-high: $0.4 < \cos\theta < 0.7$; medium-low: $0.14 < \cos\theta < 0.4$; low: $\cos\theta < 0.14$). For each list, 8 pairs of words with high semantic similarity were selected. Next, words within the lists were organized such that members of a pair did not appear adjacent to one another. In addition, no adjacent pair had a high similarity. This list construction method was used in previous studies[45], as well as in the pilot study. For each list, four different ordering schemes of the words within the list were created. The ordering schemes were counterbalanced across participants, with random assignment to one of the schemes.

### Justifications
Participants' introspective reflections regarding the information that triggered recall, or "justifications", were self-reported. Instructions before the justification-recall phase were: "You will now be shown again the words you just remembered. Please explain in as much detail as possible why you think this particular word appeared in the study phase. Consider any factors that you think might justify your recollection of the word". During the justification-recall phase participants were shown one word at a time produced by the participant during the item-recall phase and probed with the instructions: "Please describe in as much detail as possible why you believe the word above appeared in the study phase".

The procedure, used in the pilot study, was inspired by previous studies in which such justifications were collected[38,40]. Following each free recall test phase, participants were asked to describe the information they used to recall the words. Participants were cued with each word recalled during the test phase, and justifications were typed by participants. The justifications were manually coded for analysis into one of two exclusive groups: with-justification (J + ) (i.e., an item recalled with a specific description such as: "I recalled the pearls because I made up a story about a pearl necklace"), or without-justification (J-) (i.e., an item recalled without an informative description, such as: "I don't know why, I just remembered it"). The inter-rater reliability was measured to ensure a strong agreement between manual raters, the achieved reliability calculated by a Cohen's Kappa statistic was $\kappa = 0.89$ (95% CI: 0.85-0.92), thus maintaining the pre-registered criterion of Cohen's Kappa of 0.8[74].

### Justifications' detail and abstraction
To further explore the effects of time on justifications, each justification was analyzed for its number of internal/external details. This analysis is based on the Autobiographical Interview[62,63], in which the number of details providing distinct pieces of information are counted, and then classified as internal or external to the event described. The analysis was performed using the Automated Autobiographical Interview Scoring tool[64], a Neural Network based model for automated classification of internal and external

details in a text (A Google Colab Notebook and instructions for the automated scoring can be found at: https://github.com/rubenvangenugten/).

In addition, justifications' average level of concreteness/abstraction were assessed, because abstraction may be reflective of transformation from a contextually-detailed memory to gist-based memory[75,76]. Justifications' concreteness levels were quantified using the concreteness norms for individual words[65], and multiword expressions[66].

Word frequencies[69], and word keyness scores[70] were calculated for the justifications at each time delay to assess if there were shifts in linguistic content as evidenced by differential word prevalence in justifications of each delay condition.

### Experimental procedure

Participants recruited from Prolific were directed to the secured study website, where they provided informed consent. They were then randomly assigned to one of two delay order conditions: a delay of one day between study and test phases in the first list followed by a delay of 1.5 minutes for the second list or vice versa. The one day interval was aimed at examining an ecological delay. Namely, one which corresponds to delays at which there is a considerable decrease in everyday occurrences recalled[77]. It has recently been argued that memory for events after delays of such timescales (12 h -7 days) is a specific stage of Long Term Memory ("Transitional Long Term Memory"[78,79]). Recall at such a delay has also been tested in Pilot study B (See Supplementary Note 2 | Pilot study 2 + 3). Participants were restricted to performing all phases of the study on a desktop computer (excluding smartphones and tablets).

The experiment consisted of a total of two blocks. Each block includes four stages: I. study phase, II. Time Interval / Distraction task, III. Free recall, IV. Self-report of justifications. Figure 1 illustrates the experimental sequence.

**Study phase**. In each study phase, each of the 16 words were presented in the middle of the screen for 4000-ms, followed by a fixation cross which appeared on the screen for 1000-ms. Participants were requested to remember as many words as possible.

**Distraction task**. In the 1.5 min delay conditions, during the delay between the study and free recall phases, participants solved math problems of the form $X + Y + Z = ?$ (where X,Y,Z are single digit integers) between study and test. This was chosen as a distraction task as it is engaging enough as to minimize rehearsal and is not verbal, therefore should not interfere with mnemonic processing of the word lists[31].

In the one day delay conditions, after the study phase, participants received instructions to revisit the experiment website after one day for a test phase and received a reminder by email.

The order of the delay conditions (Short/Long) was counterbalanced across participants. Participants who started with the Short-Delay condition, performed the Short-Delay study, distraction task, and test, and then immediately the Long-Delay study on the first day. They then returned after a long delay (one day) to perform the Long-Delay test. Participants who started with the Long-Delay condition, performed the Long-Delay study on the first day and returned after a long delay (one day) to perform the Long-Delay test, and then immediately performed the Short-Delay study, distraction task, and Short-Delay test.

**Free recall test phase**. Immediately following the distraction task or upon revisiting the website after the time interval, participants were requested to type in words which they recalled from the study phase, one word at a time in any order. Time for each item was unlimited, but the entire test phase was limited to two minutes.

**Justifications recall phase**. Following the free recall test phase, the recalled words were presented individually in the order in which they were recalled by the participant (both correct and incorrect recalls were presented). For each of the presented words, participants were instructed to describe the information they used to recall that particular word (Fig. 1). The screen for typing each justification was displayed until participants pressed 'Enter', or until the limit of 30 s per word elapsed. After this, participants were asked to rate their confidence in their recall, regardless of whether a justification was provided, on a scale from 1 to 6 justification.

**Post-experiment questionnaire**. Following completion of the experiment, participants were asked a few questions regarding the manner in which they performed the various stages of the experiment. Participants were asked whether during the distraction task they rehearsed studied words or focused on solving the equations. Additional simple attention checks were administered here (see exclusion criteria).

### Sampling plan

**Bayesian analyses**. To assess the effects of delay duration on level of detail (Hypothesis 3), Bayesian t-tests were conducted. The sampling plan for these tests was the collection of a sample size required for a Bayes Factor score of 10 in support of the null hypothesis, or a comparable size in support of the differing hypothesis, or until reaching the constraint of $N = 500$ participants). We hypothesized that the levels of detail of justifications (for items recalled with justifications) will either decline meaningfully between short and long delays, supporting gradual forgetting of contextual information or will remain similar after both long and short time delays. supporting preservation of contextual information (an all-or-none pattern of forgetting). Conducting paired t-tests for levels of detail after each time delay, a Bayes Factor of 10 or greater in support of the null hypothesis ($BF_{01} > 10$) was taken as credible evidence in favor of the all-or-none account. A gradual form of forgetting, on the other hand, would be demonstrated by a Bayes Factor of 10 or greater in support of the differing hypothesis ($BF_{10} > 10$).

We conducted a simulation analysis for the other hypotheses to determine the degree of evidence achieved in the Bayesian tests used for the other hypotheses included in the study. We sought to determine whether our sampling plan targeting Hypothesis 3 is sufficiently sensitive so as to provide conclusive results for the *other* hypotheses. Using the results of Pilot Study 2 (Supplementary Note 2 | Pilot study 2 + 3), responses for 500 participants were simulated over 200 iterations. These provided a sample of 200 Bayes Factors (one per sampling iteration), which we used to calculate 95% confidence intervals for the predicted Bayes Factor achieved. The resulting confidence interval ranges are shown in Table 2. Hypothesis 1 and Hypothesis 2 demonstrated sufficient evidence achieved (i.e., 95% CIs demonstrated Bayes Factors greater than 10). Hence, H1 and H2 were included as confirmatory analyses. An additional three hypotheses (E1-E3) which did not demonstrate definitive evidence to one direction, were included as pre-registered exploratory analyses.

### Exclusion criteria

To filter out automated answering bots and inattentive participants, attention checks were combined in the study procedure. Specifically, while presenting the instructions before the study phase, participants were required to briefly repeat the instructions in their own words. This was done by typing in a text box at the bottom of the instructions' page. The answers were screened manually, and responses containing only nonsense words or characters and no coherent words, or containing only text irrelevant to the experiment (e.g., text copied from a Wikipedia page) will lead to participant exclusion. Justifications were screened for exclusion (as well as classification) by two separate raters. To ensure consistency in exclusion criteria, the Inter-rater reliability was assessed and achieved reliability was Cohen's Kappa of $\kappa = 0.83$ (95% CI: 0.65–1), thus achieving the pre-registered standard of Cohen's Kappa of at least 0.8.

In addition, simple attention checks were included (e.g., correctly identifying a picture of a cookie, as "someone's dessert", from a

**Table 2 | Simulated Bayes Factor sizes for the additional hypotheses**

| Hypothesis | Bayes Factor 95% Confidence Intervals |
|---|---|
| H1. Higher recall accuracy will be found for items recalled with a justification than items without a justification, supporting the claim that justifications reflect mnemonic performance. | $BF_{10}$: [6.10 ×1082 - 1.23×$10^{160}$] |
| H2. No significant interaction will be found with time delays, such that accuracy for items with justifications will not decrease over time, further supporting the validity of justifications. | $BF_{10}$: [$4.02 \times 10^{-24}$ - $1.78 \times 10^{-6}$] |
| E1. A similar proportion of items with justifications out of all items recalled will be found in the short and long delays ($BF_{01} > 10$). This will be taken as support for preservation of justifications' content over time. | $BF_{10}$: [0.05 - 0.49] |
| E2. Similar levels of lexical justifications content measures will be found in the short and long delays ($BF_{01} > 10$). This will be taken as support for preservation of justifications' content over time. Measures reflective of confidence may either decline in line with declines in subjective confidence, or may remain intact as in other justifications' content, implying an implicit association with maintained justification content. | Concreteness: $BF_{10}$: [2.40 ×108 1.14 ×1019] Tentativeness: $BF_{10}$: [0.06 - 5.13 ×104] Certitude: $BF_{10}$: [0.05 - 0.38] |
| E3. Similar levels of justifications' unigram frequencies will be found in the short and long delays ($BF_{01} > 10$). This will be taken as support for preservation of justifications' content over time. | $BF_{10}$: [0.24 - 8.15 ×104] |

**Table 3 | Recall Accuracy by Delay (Short/Long) and Justification categories (J-/J + )**

| Predictors | M0 Odds Ratios | CI | p | M1 Odds Ratios | CI | p | M2 Odds Ratios | CI | p |
|---|---|---|---|---|---|---|---|---|---|
| Intercept | 45.99 | 34.64-61.06 | **<0.001** | 20.15 | 14.87-27.30 | **<0.001** | 19.95 | 14.19-28.06 | **<0.001** |
| Delay [Long] | 0.05 | 0.04-0.06 | **<0.001** | 0.05 | 0.04-0.06 | **<0.001** | 0.05 | 0.03-0.07 | **<0.001** |
| Justification [J + ] | | | | 3.79 | 2.94-4.89 | **<0.001** | 3.86 | 2.62-5.69 | **<0.001** |
| Delay × Justification | | | | | | | 0.97 | 0.61-1.54 | 0.905 |
| **Random Effects** | | | | | | | | | |
| $\sigma^2$ | 3.29 | | | 3.29 | | | 3.29 | | |
| $\tau_{00}$ | 3.11 Participants | | | 3.02 Participants | | | 3.02 Participants | | |
| ICC | 0.49 | | | 0.48 | | | 0.48 | | |
| N | 421 Participants | | | 421 Participants | | | 421 Participants | | |
| Observations | 5335 | | | 5335 | | | 5335 | | |
| Marginal $R^2$ / Conditional $R^2$ | 0.251 / 0.615 | | | 0.291 / 0.630 | | | 0.291 / 0.630 | | |

Predictor estimates in the Binomial GLMM are displayed here on the Odds-Ratio scale. The CI column displays 95% Confidence Intervals for the Odds-Ratio estimates.
Bolded values indicate statistically significant effects.

few distractors). Failure to answer the correct choice on these questions will also serve as an exclusion criterion. These measures were meant for filtering out automated text generation bots, and completely inattentive participants.

Data-driven criteria for exclusion will be recall of no items per list, accuracy of less than 50% on items recalled (i.e., more than half of the words produced in the test phase are incorrect recalls), and providing justifications for fewer than half of the items probed.

Study recruitment will continue until 500 participants are collected who meet the inclusion criteria. Sampling plan is summarized in Table 1.

Data and statistical analyses were performed with R version 4.4.0[80]. Generalized linear mixed model (GLMMs) were conducted with the lme4 package[81], and post-hoc analyses were conducted with the emmeans package[82]. Bayesian t-tests and Bayesian Binomial tests were conducted with the BayesFactor package[83].

Data preprocessing included extracting participant responses on the test phase, and running them through a spellchecker, correcting misspelled responses, using the Hunspell package[84]. Items presented at study were indexed by their presentation order, and so were items recalled at test, to evaluate correct/incorrect recalls, and effects pertaining to the order of retrieval.

### Generalized Linear Mixed Models (GLMMs)
To overcome the unbalanced numbers of items with vs. without justifications, we conducted the analyses of recall metrics comparing items with and without justifications using mixed models. In line with recommendations for analyzing accuracy scores with a logistic mixed model[85], recall accuracy was examined with a logistic mixed model.

### Bayesian analyses
Bayes Factors were computed using default Cauchy priors ($r = 0.707$) as implemented in the BayesFactor R package[83]. This prior specification follows standard recommendations for behavioral data and allows direct comparison to previous memory studies using the same approach. For all Bayesian t-tests and model comparisons, we report the Bayes Factor ($BF_{10}$ or $BF_{01}$) as a measure of evidence for the alternative or null model, respectively. Posterior distributions were summarized by their mean and 95% credible intervals.

### Confirmatory analyses
**Is there delay-dependent forgetting?** To confirm the manipulation of forgetting over time, we assessed the number of correctly recalled items after a short delay (M = 7.41, SD = 3.54) and after a long delay (M = 3.18, SD = 3.41). A paired t-test demonstrated a significant decrease in the number of items correctly recalled from the short to the long delay ($t(420) = 22.72$, Cohen's $d = 1.11$, $p < 0.001$, $BF_{10} = 2.39 \times 1071$), confirming the effect of delay-dependent forgetting.

**Do justifications reflect mnemonic performance?** To affirm the validity of memory justifications in subjective reporting of memory performance, we compared the likelihood of correctly recalling items with and without justifications. To this end, we fitted a series of generalized mixed models with binomial outcomes.

We first fitted a baseline model (M0: -2LL = 3314.1, BIC = 3339.8; see Table 3) including the fixed effect of delay and the random effects of study participants' intercepts. We then fitted our first model (M1: -2LL = 3203.9, BIC = 3238.2), which added to the baseline model the fixed main effect of

**Fig. 2 | Accuracy of recall by delay and justification.** Proportion of correct recalls as a function of justification type (Justification / No justification) and delay (Short / Long). Error bars represent 95% confidence intervals; *n* = 421 participants. *Orange bars represent items recalled with justifications; blue bars represent items recalled without justifications.*

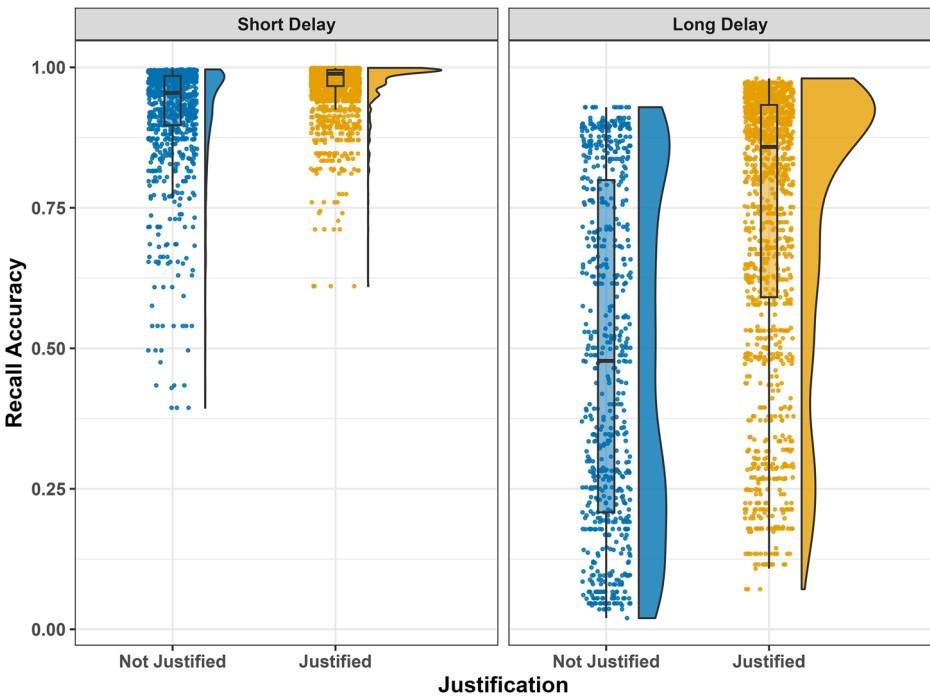

justifications (items with and without justifications). Model comparison showed that M1 demonstrated a significantly better fit than M0 ($\chi^2(1) = 110.19$, $p < 001$, $BF_{10} = 1.16 \times 10^{22}$), confirming that self-reported justifications are associated with mnemonic performance.

In M1, the main effect of Delay was significant, demonstrating the effect of delay-dependent forgetting (*Odds Ratios* = 0.05, $p < 0.001$). The main effect of justifications was significant as well (*Odds Ratios* = 3.79, $p < 0.001$). Follow-up analyses demonstrated that items with justifications (J + % correct: 0.94) had a significantly greater proportion of items correctly recalled than items without justifications (J- % correct: 0.82 ; Z = 10.27, $p < 0.001$; See Fig. 2). These results suggest self-reported memory justifications are a valid reflection of mnemonic performance.

**Is the link between justifications and recall accuracy affected by study-test delay?** To assess whether the link between memory justifications and memory performance is mediated by time delay we fitted an additional model, adding to M1 the Delay X Justification interaction (M2: -2LL = 3203.8, BIC = 3246.8). Model comparison demonstrated that M2 did not improve model fit over M1 ($\chi^2(1) = 0.01$, $p = 0.905$, $BF_{10} = 0.01$). A Bayesian analysis strongly favored the simpler model (M1), with a Bayes Factor of $BF_{01} = 72.5$. These results demonstrate no interaction effect between delay and justifications for recall accuracy, indicating that the justifications were equally diagnostic of accuracy across both delays, consistent with the pre-registered prediction. However, a follow-up analysis focusing on justified items only revealed that the proportion of justified items that were correct significantly declined from the short-delay (M = 0.99) to the long-delay condition (M = 0.82) (t(306) = 11.52, $p < 0.001$, $d_a = 0.66$, $BF_{10} = 3.7 \times 10^{22}$).

**Is justifications' content preserved over time?** Our main study hypothesis revolved around the effect of time delay on the contents of justifications. To assess justifications' level of detail, we counted the proportion of episodic details in participants' justifications for correct recalls using the automated autobiographical interview tool (van Genugten & Schacter, 2024). The proportion of episodic details after the long delay (M = 0.486, SD = 0.3), and after the short delay (M = 0.485, SD = 0.3) did not differ significantly (t(247) = − 0.04, p = 0.97, *Cohen's d* = 0.002, $BF_{10} = 0.071$). A Bayesian analysis demonstrated strong

support for the null hypothesis ($BF_{01} = 14.06$) confirming no difference in justifications' level of detail between the short and long delays (See Fig. 3). These results suggest the contents of recall justifications remain preserved over time.

**Pre-registered exploratory analyses**
**Are the proportions of justifications steady over time?** To examine whether the proportion of correct recalls that included justifications varied with time delay, we calculated, for each participant, the proportion of correctly recalled items that were accompanied by a justification, for both the short and long delay conditions. The average proportion was nearly identical across delays: M = 0.71 (SD = 0.35) for the short delay and M = 0.72 (SD = 0.38) for the long delay (t(313) = 0.16, *Cohen's d* = 0.009, p = 0.869, $BF_{10} = 0.06$). A Bayesian analysis yielded strong evidence in favor of the null hypothesis ($BF_{01} = 15.59$), suggesting that the proportion of items with a justification did not differ between delay conditions.

**Are Justifications' contents, as reflected in lexical measures (concreteness, tentativeness, certainty), similar across time delays?** To examine whether lexical measures of justifications varied with time delay, we compared justifications' values of concreteness (i.e., use of terms with high concreteness norms, e.g., "tulip", "armchair", versus terms with low concreteness e.g., "belief", "essentialness"), tentativeness (use of tentative terms, e.g., "if", "or", "any", "something"), and certainty (e.g., "really", "actually", "of course") between the Short-delay and Long-delay conditions using Bayesian paired t-tests.

Concreteness values, derived from a single-word norms dictionary[65], were significantly higher in the Short-delay condition (M = 2.62, SD = 0.28) than in the Long-delay condition (M = 2.56, SD = 0.24; t(247) = 3.37, *Cohen's d* = 0.21, p < 0.001). A Bayesian analysis demonstrated strong support for the alternative hypothesis ($BF_{10} = 17.32$), suggesting justifications' concreteness varied by time delays. We additionally applied a recently developed norms dictionary, which provides concreteness ratings for multi-word expressions[66]. When concreteness was calculated from the multi-word norms, no significant difference was found between Short-delay (M = 2.55, SD = 0.46) and Long-delay (M = 2.53, SD = 0.51; t(162) = 0.36, *Cohen's d* = 0.03, p = 0.72, $BF_{10} = 0.09$) conditions. A Bayesian analysis

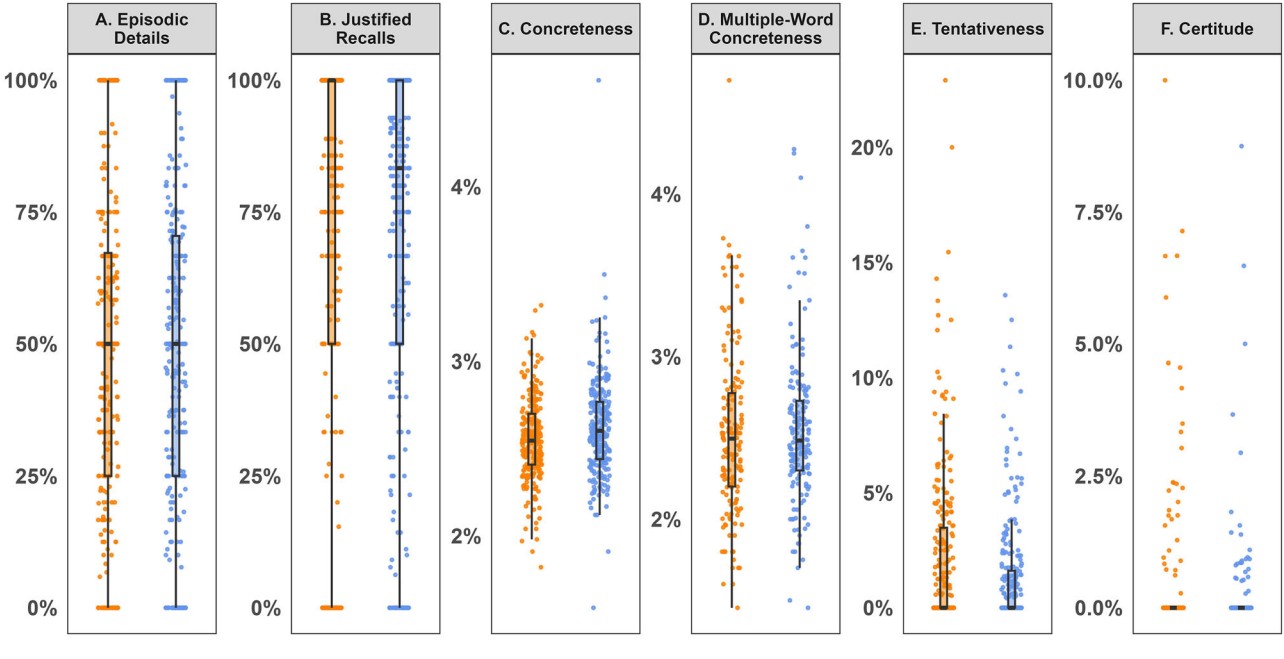

**Fig. 3 | Justification contents by delay.** Justifications' level of detail (**A**), proportion of justified recalls (**B**), and lexical measures of justifications (**C**–**F**) across Short- and Long-delay conditions. Error bars represent 95% confidence intervals; $n = 421$ participants. *Blue bars represent the Short-delay condition; orange bars represent the Long-delay condition.*

demonstrated strong support for the null hypothesis ($BF_{01} = 10.76$), indicating justifications' concreteness did not vary by time delays. These results show that delay-related changes in concreteness emerge when words are scored individually, but disappear when the analysis is based on multi-word expressions.

Justifications' tentativeness scores were significantly lower in the Short-delay condition ($M = 1.32$, $SD = 2.42$) than the Long-delay condition ($M = 2.13$, $SD = 3.54$; $t(247) = -3.20$, *Cohen's d* $= 0.20$, $p = .002$) conditions. A Bayesian analysis demonstrated strong support for the alternative hypothesis ($BF_{10} = 10.08$), suggesting justifications' tentativeness varied by time delays.

Finally, justifications' certainty scores were significantly lower in the Short-delay condition ($M = 0.19$, $SD = 0.85$) than the Long-delay condition ($M = 0.36$, $SD = 1.25$; $t(247) = -2.40$, *Cohen's d* $= 0.15$, $p = 0.017$) conditions. A Bayesian analysis, however, demonstrated only anecdotal support for the alternative hypothesis ($BF_{10} = 1.17$), indicating an inconclusive difference in justifications' certainty between delay conditions.

**Are justification contents, reflected in unigram frequencies, similar across time delays?** To examine whether participants used different language when justifying correctly recalled items across delay conditions, we compared unigram (single-word) frequencies between the Short and Long conditions.

Unigram frequencies were computed, and rare terms (appearing fewer than five times) were excluded. Unigram distributions were compared using a Bayesian Wilcoxon signed-rank test.

No significant difference was found in word usage between conditions ($V = 8.94 \times 10^4$, $p = .712$, $BF_{01} = 0.05$), with a Bayesian analysis showing strong support for the null hypothesis of no difference between conditions ($BF_{10} = 20.48$). This result suggests that participants employed fundamentally similar lexical choices when justifying their memory retrievals in the Short versus Long delay conditions.

**Non pre-registered exploratory analyses**

**Do lexical measures reflect self-reported confidence of recall?**
Following the recall of justifications, participants were instructed to designate their confidence in their correct recall of the item on a scale from 1 (very uncertain) to 6 (very certain). Confidence ratings were analyzed using a mixed-effects model with Delay (Short vs. Long), Justification (present vs. absent), and their interaction as fixed effects, and a random intercept for participant. The analysis revealed significant main effects of Delay ($F(1, 5332) = 1285.63$, $p < 0.001$, $BF_{10} = 4.83 \times 10^{45}$) and Justification ($F(1, 5332) = 444.38$, $p < 0.001$, $BF_{10} = 1.41 \times 10^{34}$), as well as a significant Delay × Justification interaction ($F(1, 5332) = 80.94$, $p < 0.001$, $BF_{10} = 3.78 \times 10^{15}$). Pairwise comparisons showed that confidence decreased with longer delay, but this reduction was smaller for justified items (Short: $M = 5.74$, 95% CI [5.65, 5.82]; Long: $M = 4.87$, 95% CI [4.77, 4.96]) than for non-justified items (Short: $M = 5.17$, 95% CI [5.07, 5.28]; Long: $M = 3.71$, 95% CI [3.60, 3.82]).

Next, we calculated pairwise correlations (per item recalled) between confidence ratings, and lexical measures derived from justifications. Numeric confidence was significantly positively correlated with justifications' concreteness, both for single word norms ($r(5301) = 0.136$, $p < 0.001$), and multi-word norms ($r(2231) = 0.086$, $p < 0.001$). Numeric Confidence was significantly negatively correlated with justifications' tentativeness ($r(5333) = -0.155$, $p < 0.001$), and to a lesser extent with justifications' certainty ($r(5333) = -0.033$, $p = 0.016$). The complete set of correlations between confidence and all linguistic and episodic measures is presented in Table 4.

We also examined whether the amount of episodic details correlated with confidence ratings. A significant positive correlation was observed ($r = 0.04$, $p = 0.004$, $BF_{10} = 1.99$), indicating evidence for a modest relationship between the proportion of episodic content and self-reported confidence.

**Table 4 | Correlations between confidence ratings and linguistic/episodic measures**

| Measure | All recall responses | Correct recalls only |
|---|---|---|
| Concreteness | $r = 0.14$ [0.11, 0.16] **$p < 0.001$, BF$_{10}$ = 8.42e + 19** | $r = 0.07$ [0.04, 0.10] **$p < 0.001$, BF$_{10}$ = 764.000** |
| Tentativeness | $r = -0.15$ [−0.18, −0.13] **$p < 0.001$, BF$_{10}$ = 6.67e + 26** | $r = -0.10$ [−0.13, −0.07] **$p < 0.001$, BF$_{10}$ = 3.38e + 08** |
| Certainty | $r = -0.03$ [−0.06, −0.01] **$p = 0.016$, BF$_{10}$ = 0.592** | $r = 0.00$ [−0.03, 0.03] $p = 0.834$, BF$_{10}$ = 0.036 |
| Episodic details | $r = 0.04$ [0.01, 0.07] **$p = 0.004$, BF$_{10}$ = 1.900** | $r = -0.00$ [−0.03, 0.03] $p = 0.935$, BF$_{10}$ = 0.035 |

Pearson's r, 95% CI, p-values, and Bayes Factors (BF$_{10}$) are shown for all recall responses and for correct recalls only.
Values are Pearson's r [95% CI]. p-values and Bayes Factors (BF$_{10}$) indicate the evidence for an association between each measure and confidence. 'All recall responses' = all recalled items; 'Correct recalls only' = correctly recalled items only.
Bolded values indicate statistically significant effects.

**Is linguistic content of justifications predictive of response accuracy, above and beyond confidence ratings?** Recent work suggests that machine learning approaches applied to linguistic features of justifications outperform traditional confidence ratings in predicting memory accuracy[30,31,41,42]. We sought to replicate this finding in our data as well. Following the methodology applied in previous studies[30,31,41,86], we trained a Bag-of-Words (BoW) classifier to predict the accuracy of recall by identifying individual words, coded by their frequency of occurrence, that are diagnostic of accurate or inaccurate recall. To assess the BoW classifier performance, and contrast it with the self-reported numeric confidence scores, we compared two receiver operating characteristics (ROCs). The BoW classifier was constructed using the log-odds predictions of the model. The three resulting ROCs are shown in Fig. 4 and the bootstrapped 95% confidence intervals demonstrate that all three predictors were reliably above chance. Receiver operating characteristic (ROC) curves for (1) the BoW language model (green); (2) Participants' reported Recall Confidence using a numeric scale 1-6 (red). Shaded ribbons indicate bootstrapped 95% confidence intervals.

The relative performance of both ROCs was tested via bootstrapping of the area under the curve (AUC) measure. The BoW classifier's prediction accuracy (AUC. 85, 95% CIs [0.83–0.87]) was superior to the numeric confidence scores (AUC.80, 95% CIs [0.80–0.84]; D = 2.40, $p = 0.016$). Thus, the BoW classifier, which is reliant on linguistic content reported within justification, outperformed the numeric confidence scores. This replicates the findings in previous studies and extends them, from recognition memory tasks to free recall.

### Reporting summary
Further information on research design is available in the Nature Portfolio Reporting Summary linked to this article.

## Discussion
The present study examined how the passage of time affects memory justifications, the introspective reflections that individuals provide in support of their recollections. Our results replicated the finding of delay-dependent forgetting[45–47], with participants recalling significantly fewer items after a one day delay compared to a 1.5 min delay. This robust effect validated our experimental manipulation. Critically, while overall recall decreased, justifications for correctly recalled items remained remarkably stable over time: the level of episodic detail, proportion of justified recalls, and word usage patterns did not differ between short and long delays, supporting an all-or-none pattern of forgetting[45,55,58].

Our first hypothesis was that higher recall accuracy will be found for items recalled with a justification than items recalled without a justification. This hypothesis was strongly supported by the data: Items accompanied by justifications were significantly more likely to be correctly recalled (94%) compared to items without justifications (82%). This finding supports the claim that individuals' self-reported justifications can indeed serve as reliable indicators of memory performance. Critically, in line with our second hypothesis, the validity of justifications in predicting recall accuracy (i.e., the proportion of correctly recalled items) remained generally stable over time, as indicated by the non-significant Delay × Justification interaction, $\chi^2(1) = 0.01$, p = .905, BF$_{10}$ = 0.01 (BF$_{01}$ = 72.5). However, a follow-up analysis restricted to justified items revealed that the proportion of justified items that were correct significantly declined from the short-delay to the long-delay condition. This decline parallels the decrease in confidence ratings over time and is in line with a general decline in accuracy, rather than the diagnostic value of justifications.

Thus, justifications, if they are present, reflect stable features of retrieval, rather than transient cues that degrade over time, aligning with theoretical accounts of recollection as a robust authentication mechanism[30,34,87]. When individuals can produce justifications for their memories, regardless of delay length, the justifications remain valid markers of genuine memory retrieval.

Our core hypothesis addressed the question of whether justifications' content is preserved over time. Here we contrasted between the predictions of two differing theoretical accounts. The gradual degradation account[51,52] predicted a decline in justifications' level of detail, supporting a fragmenting in contextual information over time. In contrast, the all-or-none forgetting account[45,55,58] predicted a similar level of detail across time delays, demonstrating preserved contextual information over time. Our analysis using the automated autobiographical interview tool revealed no difference in episodic detail of justifications for correct recalls between short and long delay conditions (BF$_{01}$ = 14.06). This finding supports an all-or-none account for forgetting of contextual information. Similarly, we found that the proportions of items recalled along with justification remained similar across time delays. Though there are fewer correct recalls after long delays, items nonetheless correctly recalled demonstrate similar levels of episodic detail and similar proportions of justified recalls, in line with the all-or-none forgetting account. Finally, word usage patterns as assessed by unigram frequency, remained remarkably similar following a delay.

Our findings provide insights into the specific mechanism that may underlie forgetting. The fact that episodic detail, word frequency and the proportion of justified items were all preserved for correctly recalled items suggests that the underlying memory traces remained intact and rich. Our results are in line with theoretical models of episodic retrieval based on threshold-like access processes. According to these models, successful recollection occurs only when internally generated retrieval cues reach a critical threshold of activation, enabling access to intact memory traces. When context reinstatement fails, the activation may fall below threshold, resulting in all-or-none forgetting (e.g., refs. 87,88). This framework offers a plausible explanation for our finding that the level of episodic detail in justifications did not gradually decline over time but rather remained stable for the subset of items that were successfully recalled, supporting an all-or-none pattern of contextual memory retrieval.

Additional linguistic analyses demonstrated a more complex picture. Justifications provided after a longer delay included more tentative expressions (e.g., "if", "or", "any", "something"). Single word concreteness scores decreased from short to long delays, while multiple word concreteness scores did not differ between time delays. These lexical patterns are in line with prior research suggesting that language use reflects underlying cognitive and metacognitive processes, such as monitoring memory sources and evaluating the plausibility of recollections[18,19]. Justifications' certainty expressions (e.g., "really", "actually", "of course") were low (less than 0.5%) in both conditions and showed merely anecdotal differences between conditions. Therefore, it is difficult to draw strong conclusions from this result. It is possible that our analysis was underpowered for some of the lexical analyses, namely, certainty and multiple word concreteness. Most justifications were short and concise, hence far fewer multiple word expressions were to be found in the justifications. Future studies with longer

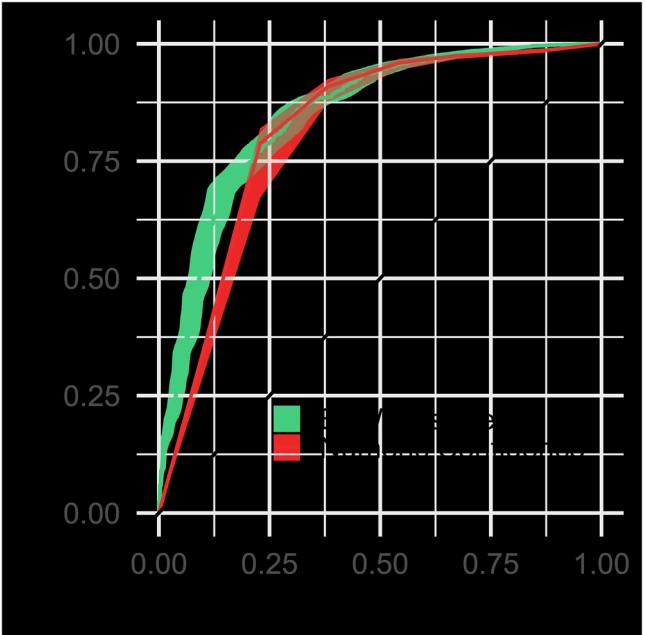

**Fig. 4 | Accuracy of justification classification models.** Receiver operating characteristic (ROC) curves comparing prediction accuracy for recall correctness based on linguistic features of written justifications (green) and numeric confidence ratings (red). Shaded ribbons show bootstrapped 95% CIs.

and more varied justifications or using big-data sources may produce a more comprehensive description of these lexical measures.

Taken together, these lexical differences suggest that while time does not affect the amount of detail used when expressing and validating memories, it might induce a reduction in the concreteness of the accessible justification information over time. Given the strong association between delay and confidence, and between confidence and linguistic markers of tentativeness and one measure of concreteness, it is plausible that the effects of time on some language measures are mediated by decreases in confidence. Indeed, participants' numeric confidence ratings declined following a delay, and linguistic markers of tentativeness increased, even though the overall proportion of justified items and the level of episodic detail in the justifications remained stable. In addition, the proportion of justified items that were correct declined, consistent with the decrease in confidence ratings over time.

Recent work suggests that machine learning approaches applied to linguistic features of justifications outperform traditional confidence ratings in predicting memory accuracy[30,31,41,42]. Our findings extend this notion to delayed retrieval, highlighting that the content of justifications, particularly elements tied to the remembered event, may offer a temporally stable window into episodic memory, beyond what is captured by subjective confidence. In legal or forensic contexts, where testimony often involves delayed recollection, encouraging witnesses to produce written justifications may provide a richer and more reliable basis for evaluating memory validity.

Our findings establish self-reported memory justifications as a useful research tool for investigating the dynamics of episodic memory. Memory justifications not only reflect objective mnemonic performance (i.e., recall accuracy), but also align with subjective confidence ratings, as indicated by the finding that recall confidence scores were correlated with justifications' tentativeness, certainty and concreteness. These results complement prior work, which has shown that confidence judgments and metacognitive monitoring tend to deteriorate over time, leading to reduced calibration between confidence and accuracy[53,54]. While some metacognitive indicators, such as willingness to report, may still retain partial diagnostic value even after extended delays[54], our findings suggest that numeric confidence ratings alone may be insufficient. In contrast, language-based justifications appear

to better retain their diagnostic utility over time, offering a more temporally stable and content sensitive window into the quality of episodic memory.

Beyond their mnemonic value, justifications also hold theoretical significance as reflections of epistemic self-monitoring. Studies analyzing individuals' memory justifications may illuminate the interplay between objective memory performance and subjective experience. It has been proposed that memory justification operates as a form of epistemic vigilance, a cognitive mechanism essential for evaluating the reliability of information communicated by others[5,6,86] Our findings demonstrate that this evaluative capacity extends beyond socially shared memories: individuals also recruit justification processes to assess the validity of their own recollections[8]. In this sense, justification reflects an internalization of the epistemic standards applied in communication, now turned inward. Future research may explore how social frameworks surrounding memory communication shape both mnemonic and metamemory processes[8,89]. Verbal memory justifications present a promising tool for these investigations owing to their dual objective and subjective nature.

## Limitations

First, an alternative account for the specific mechanism underlying forgetting remains plausible. Stable justification fidelity over time could, in principle, also arise from a graded degradation of contextual features that affects accessibility and fidelity in tandem. This alternate account would explain the current findings due to the content of some contextual information prone to degradation over time, whereas other information remains stable over time. The items whose contextual information degraded to a certain amount may have been inaccessible. In contrast, items whose contextual information which did not degrade remained accessible and with maintained levels of fidelity. The current study cannot elucidate between these two accounts because there are no justifications for items which were not accessible. This could, however, be elucidated in future studies where the recall test would be followed by a recognition test for all items that appeared in the list including those that were not accessible in recall. Most importantly, however, according to both accounts, the content and quality of justifications for items which were accessible during recall remains intact. Thus, justifications remain a viable reflection of memory accuracy even after delay.

Apart from this theoretical limitation, there are some methodological details that warrant discussion. Our study offers an initial step in combining research on participants' introspective memory justifications alongside objective mnemonic performance in a free recall test for word lists. While even "simple" stimuli, such as single words, elicited a wide range of responses and personal associations, participants typically provided short responses, as they were asked to justify their word recalls. Future research may explore richer and more complex episodic memories, such as eyewitness accounts or narratives recounting video clips, to better capture the multifaceted nature of real-life experiences. In such contexts, each recalled element within a narrative could be viewed as functionally analogous to a single recalled word in the present paradigm representing a discrete unit of episodic information that a justification can accompany. Namely, justifications will accompany individual remembered details, rather than entire events, such that some components of a complex episode might be recalled with diagnostic justifications while others might be forgotten or lack justifications. Examination of such naturalistic memories, including multiple elements, may offer deeper insights into how complex events are represented in memory and how individuals justify such recollections. Furthermore, future studies may benefit from eliciting justifications through additional modalities (e.g., collecting video recordings of participants giving memory justifications), as human communication relies not only on semantic content but also on auditory (e.g., voice tone) and visual (e.g., gesture, facial expression) cues. Such extensions may further clarify the extent to which memory justifications generalize beyond controlled settings.

Finally, our starting point was that recollection is an adaptation meant to address the problem of distinguishing between veridical and non-veridical memories[8]. Thus, recollective content provides a justification for

the veracity of memories. This rationale inspired our use of subjective memory justifications and the measurement of episodic detail within them. Still, other theorists emphasize the role of recollection in reconstructions, and flexibly recombining episodic details, not only in memory retrieval, but also in simulating possible future events[32–34]. Regardless of the reason for the emergence of recollection during evolution, it nevertheless provides diagnostic information about memory's accuracy.

In sum, the present study demonstrated that memory justifications serve as robust indicators of recall accuracy. This pattern holds true following a delay, highlighting their value for both theoretical inquiry and the assessment of practical credibility. The prevalence of justifications, their association with objective memory performance, the distribution of lexical features, and the richness of episodic detail all remained stable over time. These findings suggest that although access to contextual information may diminish over time, the fidelity of that information, once retrieved, remains preserved, supporting an all-or-none pattern of forgetting. By integrating self-reported justifications with objective memory measures, this study offers insights into the cognitive and metacognitive mechanisms underlying memory validation.

## Data availability

All study materials (e.g., word lists) and all behavioral data collected via the online study were anonymized and are freely available at the Open Science Framework (OSF) website (https://osf.io/3wu5a/). The pilot study data are freely available at the same location.

## Code availability

All R scripts for analysis of this study are available on the OSF website (https://osf.io/3wu5a/). The R scripts used to analyze the pilot study and run the power analysis for the study are freely available at the same location.

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

## Acknowledgements

This research was supported by THE ISRAEL SCIENCE FOUNDATION, grant No. 2055/22 to T. S. and grant No. 1324/23 to M.G. The funders have/had no role in study design, data collection and analysis, decision to publish or preparation of the manuscript. The authors thank Alon Scheuer, for his invaluable contribution, particularly in building the study's experiment.

## Author contributions

A.G., M.G., and T.S. conceptualized the study. A.G. designed the experiment. A.G. and Z.R.G. curated and analyzed the data. M.G. and T.S. obtained funding. A.G., Z.R.G., M.G., and T.S. wrote the manuscript. All authors reviewed and approved the final version of the manuscript.

## Competing interests

The authors declare no competing interests.
