## [Transparent Peer Review file · Communications Psychology]

Memory justifications provide valid indicators of retrieval accuracy across time

Corresponding Author: Ms Zohar Raz Groman

Version 0:

Decision Letter:

Dear Mr Gamoran,

Thank you once again for your manuscript, entitled "Why we remember now and later: examining memory justifications over time," and for your patience during the peer review process.

I am sorry for the delay in returning to you with a decision, which came about as a result of the combination of an initial delay in soliciting reports from experts in your field, paired with a later medical absence.

Your manuscript has now been evaluated by 3 reviewers, whose comments are included at the end of this letter. Although the reviewers are enthusiastic about the research question, but they also raise some serious concerns. We are interested in the possibility of proceeding further with your submission in Communications Psychology, but would like to consider your response to these concerns in the form of a revised manuscript.

You will find that the reviewers criticise the strength of theoretical support for your hypotheses (or the appropriate embedding in the literature), and the mapping between hypotheses and implementation. In addition, they raise questions about some analysis choices and details of the protocol (statistics, exclusion criteria), as well as including suggestions to improve the presentation and potentially add further analysis.

The amount of work that is necessary in revision is substantive and if you decide to rather seek publication elsewhere, we would be grateful if you could let us know.

However, if you intend to resubmit a revised version of the protocol, please focus on the following key aspects as you address all reviewer concerns.

1) Conceptual clarity and matching operationalization

Registered reports present confirmatory research, which means that the results, positive, negative, or null, must inform a clear research question; this question may be derived from a strong theoretical framework, or consist of a replication of published work. Your project falls into the former category. Please ensure that your revision contains a revised or clarified set of hypotheses that are clearly derived from the literature, so that the results obtained through the research are informative for the field. Please review your dependent variables and how these map on to the hypotheses. This includes the question posed to the participants to gauge the degree of "justification". Editorially, we ask you to pay special attention to concerns about task-order, or "priming" effects.

2) The use of pilot data

Any changes to the paradigm at this stage should be accompanied by new pilot data. Registered Reports receive in-principle acceptance before the research project commences. However, in exceptional cases, Stage 2 protocols are still rejected, because it becomes apparent that the data generated by the paradigm are unsuitable, or basic manipulation checks fail. This should be avoided this at all costs. For this reason, we ask authors of new paradigms to provide feasibility pilot data. Addressing the referees' concerns (especially Reviewer #2) will require changes to the design and we ask you to run a small scale study in which you test the appropriateness of your protocol, for example with regard to participants' responses to the instruction/question, exclusion criteria, attention checks, participant attrition, and the suitability of the data for the planned analysis (for example, do participants show floor/ceiling effects?). This undertaking will also allow you to deposit your analysis pipeline (code) a priori to commencing the actual research. Please note that while the existing (and

additional) pilot data offers a general sense of the effects, pilot data on its own is not a suitable basis to derive effect size estimates for power analyses.

3) Power analysis / statistics

The protocol should contain only one sampling strategy. There are two options.

i) You may either conduct a power analysis to determine the size of the sample that will provide a priori power of 0.95 or higher for all proposed null-hypothesis significance tests. In other words, whichever one of your hypotheses has the smallest targeted effect size determines the ultimate sample size. The target effect size must be chosen as the smallest effect size of theoretic or pragmatic relevance and a suitable justification must be included. Target effect sizes cannot be derived solely from a pilot study or a single published study (using a number of studies reporting comparable effects as a basis can be helpful).

ii) You may alternatively opt for Bayesian sampling, where you commit to continue sampling until a $BF > 10$ either in favour of the H_0 or the respective hypothesis is obtained. In this case, you must determine which of your hypotheses is the key hypothesis that the protocol is determined to answer and this will drive the final sample. The evidence for or against other hypotheses could remain inconclusive as a result. If you face resource limitations, you are permitted to specify a maximum feasible sample size at which data collection must cease regardless of the Bayes factor; however to be eligible for advance acceptance this number must be sufficiently large that inconclusive results at this sample size would nevertheless be an important message for the field.

Please see also: <https://www.nature.com/commpsychol/submit/registered-reports> for more guidance on sampling strategy, power analysis, and statistics.

In sum, we invite you to revise your Stage 1 Registered Report taking into account reviewer and editor comments. Please highlight all changes in the manuscript text file.

* Include a "Response to reviewers" document detailing, point-by-point, how you addressed each referee comment. If no action was taken to address a point, you must provide a compelling argument. This response will be sent back to the reviewers along with the revised manuscript.

* Ensure that you use our template for Stage 1 Registered Reports to prepare your revised manuscript:

https://www.nature.com/documents/NHB_Template_RR_Stage1.docx

Failure to ensure that your revised Stage 1 submission meets our requirements as specified in the template will result in your submission being returned to you, which will delay its consideration.

* In your cover letter, please include the following information:

--An anticipated timeline for completing the study if your Stage 1 submission is accepted in principle.

--A statement confirming that you agree to share your raw data, any digital study materials, computer code, and laboratory log for all eventually published results.

--A statement confirming that, following Stage 1 in principle acceptance, you agree to register your approved protocol on the Open Science Framework (<https://osf.io/>) or other recognised repository, either publicly or under private embargo, until submission of the Stage 2 manuscript.

--A statement confirming that if you later withdraw your paper, you agree to the Journal publishing a short summary of the pre-registered study under a section Withdrawn Registrations.

Link Redacted

We hope to receive your revised manuscript within eight to twelve weeks. If you cannot send it within this time, please let us know. We will be happy to consider your revision so long as the report still represents a significant contribution to the literature at that stage.

* **TRANSPARENT PEER REVIEW:** Communications Psychology uses a transparent peer review system. This means that we publish the editorial decision letters including Reviewers' comments to the authors and the author rebuttal letters online as a supplementary peer review file. We publish these records for all accepted manuscripts. However, on author request,

confidential information and data can be removed from the published reviewer reports and rebuttal letters prior to publication. If your manuscript has been previously reviewed at another journal, those Reviewers' comments would not form part of the published peer review file.

Communications Psychology is committed to improving transparency in authorship. As part of our efforts in this direction, we are now requesting that all authors identified as 'corresponding author' on published papers create and link their Open Researcher and Contributor Identifier (ORCID) with their account on the Manuscript Tracking System (MTS), prior to acceptance. ORCID helps the scientific community achieve unambiguous attribution of all scholarly contributions. You can create and link your ORCID from the home page of the MTS by clicking on 'Modify my Springer Nature account'. For more information please visit www.springernature.com/orcid.

Sincerely,

Marika Schiffer, on behalf of

Hu Chuan-Peng, PhD
Editorial Board Member
Communications Psychology
orcid.org/0000-0002-7503-5131

REVIEWERS' EXPERTISE:

Reviewer #1: cognitive psychology (memory/ meta cognition)

Reviewer #2: cognitive psychology (memory/ meta cognition)

Reviewer #3: cognitive psychology (memory/ meta cognition)

REVIEWERS' COMMENTS:

Reviewer #1:
Remarks to the Author:
See attached.

Reviewer #2:
Remarks to the Author:
Review of: Why we remember now and later: examining memory justifications over time

The proposed study has been submitted for consideration as a Registered Report (Stage 1).

General Comments

The study addresses an important and timely issue regarding the information that people use to evaluate the veracity of their memories, to "justify" those memories to themselves and to others, and whether/how the amount, quality, and validity of that information might change over time. Given that (for reasons that are still under debate) memory representations become less accessible over time, the question raised here and elsewhere, is what happens to the represented information that remains accessible? Does its quality degrade over time, or does it remain relatively preserved? If the content of accessible memories remains intact over time, then although "input-bound" forgetting in the sense of information loss may be inevitable, the "output-bound" (Koriat & Goldsmith, 1994, 1996) reliability of the information that is retrieved and reported from memory may remain relatively stable.

The proposed study is designed to examine the stability over time of a particular type of "contextual" memory content — internal thoughts and cognitive operations that occurred (and were encoded) during the processing of specific "target" items of information, and which contribute to a subjective experience of "recollection" (remembering) — if and when they are accessed together with the target information. According to various theories (most prominently, the source-monitoring framework), the accessibility (and analysis) of contextual content is used to internally evaluate and justify the veracity of the memory as a whole (including the target information), and is generally associated with high levels of both subjective confidence and actual (output-bound) accuracy.

Although many other studies (perhaps fewer than desirable) have examined changes over time in various related aspects of memory (e.g., contextual information, remember-know judgments, output-bound accuracy) and metacognitive monitoring and control (subjective confidence and its use in regulating overt memory reporting), to my knowledge the proposed study is

unique in its plan to examine change or stability in the overt verbalization of the accessible contextual content used to “justify” one’s memories. By doing so, it may potentially provide insights that go beyond what is already known on this general topic.

However, although the general theoretical motivation is clear, there are some specific theoretical issues, particularly concerning the key term “memory justification,” that require clarification and/or further elaboration. Such elaboration is also needed regarding the theoretical assumptions underlying the various experimental hypotheses, and (relatedly) what will be learned, at the theoretical level, from the expected (or unexpected) results.

1. “Memory justification”

The term is initially introduced in relation to the need for epistemic vigilance, to provide sufficient justification for believing that the information that one recalls (or recognizes) from memory is in fact veridical. According to the authors, such justification can be provided by (is one of the primary functions served by) the autoegetic/recollective content/experience that typically accompanies memory recall (and, perhaps to a lesser extent, recognition). Operationally, however, memory justifications are elicited as participants’ answers to the question “What *triggered* your recall of the item X?” (or equivalently, according to the flow diagram in Figure 1, “How did you remember the word X?”). There are two salient problems with this operationalization: (1) It does not ask, explicitly for a “justification” of why the participants believe that the recalled item is correct/veridical; rather, it asks for an introspection regarding what led to the emergence of the item into consciousness. This is assuming that the participants pay careful attention to the wording and clearly understand what is being asked. (2) As a consequence, it does not ask participants to report contextual recollective content/experience that does not serve to “trigger” the memory, but that may come to mind *after* the target item has been accessed. Such content is no less valuable in evaluating/justifying the veracity of the recall than is the content that proximally triggered the memory. (3) Particularly when the justifications are elicited in a separate phase after all items have been recalled, it seems unreasonable to believe that participants are capable of distinguishing between contextual information that may have triggered retrieval of the target, and contextual information that may have been retrieved after (via) retrieval of the target.

One might perhaps argue that participants’ inattention to the precise wording of the instructions and/or likely inability to recall the precise temporal order of the information that came to mind might in fact negate the basic concern that they are not providing all of the contextual information relevant to memory justification. At best, however, this is an unwanted source of inter-individual variance, and it would surely be preferable to reformulate the instructions so that they explicitly ask for the information that most closely corresponds to a true memory “justification.” This might be done by explicitly asking them to justify their responses (if possible) with relevant contextual memories, or alternatively, by asking them to report all of the contextual details that they can remember. The first method is of course more direct, but has the potential shortcoming of confounding the amount and quality of accessible information with the amount and quality of information that participants believe is relevant for memory justification. The latter approach avoids (as much as possible) the potential effects of such “editing,” but then the dependent variable would need to be relabeled (and conceived) as something like “verbally expressible recollective-contextual content that might potentially provide the basis for memory justification” (quite awkward), rather than memory justification per se.

2. Theoretical bases for the empirical hypotheses

The general theoretical rationale for the main experimental hypotheses was raised very briefly at the bottom of page 3, pointing (through numbered references) to findings indicating that under some conditions, and for certain types of memory content, the accessibility of the memory representations may decline over time, whereas the fidelity of the representations that remain accessible, is stable. After presenting this idea, the general research question was then formulated as “whether justifications lose fidelity over time or whether they remain quantitatively and qualitatively intact after a long time-delay.” Following up on my earlier concerns about the theoretical and operational definition of “memory justification,” let me just note that all of the various formulations suggesting that memory “justifications” are “stored” or “represented” in memory should be avoided. One may ask whether the memory justifications that people provide lose or retain fidelity over time, but it is the contextual content that provides the raw materials for such justifications, not the justifications themselves, that is (perhaps) stored and represented in memory, accessed, lost or degraded over time, and so forth. So, to paraphrase, the main question appears to be that given that a previously encoded memory can be accessed at a particular point in time, will all of the associated contextual content, encoded as part of that memory (that could potentially be used to provide a memory justification) remain intact/unchanged, regardless of the retention interval after which the memory is accessed?

As a “working hypothesis,” the authors hypothesize that the answer is “yes” — that such content and (therefore) the memory justifications that might be verbalized on the basis of that content, will remain stable over time. I use the term “working hypothesis,” because beyond the first two specific hypotheses (H1 and H2), which are rather trivial (clearly supported by “common sense” and/or existing results), the key hypothesis of “stability” in the remaining four hypotheses (H3-H6) is justified solely on the basis of the corresponding observed trends in the pilot data. No specific theoretical justifications are provided. This is problematic for several reasons. First, the conditions of the pilot study are substantially different from those of the proposed study. Beyond the number of study lists per session and (therefore) the potential role of interference, the time scales of the retention intervals differ by several orders of magnitude (1.5 minutes vs. *1 day* vs. *1 week* in the proposed study; 1.5 minutes vs. *12 minutes* in the pilot study). Indeed, although the report of the pilot study is helpful in clarifying the general idea and some of the procedural and analytical details (those that remain the same), the relevance of its results as a basis for the current hypotheses is highly questionable. Ironically, this may in fact be fortunate, because had the two studies been more similar, it would then be more problematic to treat hypotheses based on the initial pilot results as representing, in fact, *a priori* hypotheses regarding the expected results of the proposed studies.

More importantly, however, without a clear theoretical rationale for each hypothesis, it is sometimes difficult to evaluate

exactly what is being hypothesized and why, and the potential theoretical conclusions that results confirming (or refuting) each hypothesis might yield.

With regard to H3, there is an inconsistency in the precise formulation, between the text appearing on page 5 (of my version of the manuscript) and the one appearing in the summary table (Table 1). In both places it is hypothesized that the number of details per justification will remain stable over time. However, in the text, this prediction is *conditionalized* on the recalled item being correct, whereas in the summary table (and in the formulation of other similar hypotheses) it is not. Without a clear theoretical rationale for this hypothesis, it is difficult to know which version was intended.

Nonetheless, in light of the general rationale alluded to earlier, it appears to be appropriate to restrict all of the stability-related analyses to correct recalls only, in order to include only cases in which the verbalizations reflect the content of actual (and still accessible) memory representations. However, if a substantial number of verbalized justifications are provided for incorrect recalls, this would suggest that justifications can be confabulated, and if so, they might be confabulated for correctly recalled items as well (i.e., they do not necessarily reflect the content of a “retained” and accessible memory representation). This alone is not necessarily a problem, unless the tendency to confabulate/reconstruct memory justifications for correct recalls (as well as incorrect recalls) increases over time, in which case that might contribute to an “illusional” stability in quantity and quality of memory justifications over time.

H4 predicts that the proportion of recalled items accompanied by justifications will be equivalent at longer and shorter delays. Here there is no mention of conditionalizing on correct recall, but for the same reasons just discussed, it would seem appropriate to do so. Unlike H3 (and H5, H6), however, this hypothesis appears to relate to the presence or absence of recollective-contextual content (in accessible memory representations) after different retention intervals, rather than to the fidelity or quality of this content. Given the authors’ statement to the contrary (i.e., that this hypothesis relates to “quality” whereas H3 relates to “quantity”), perhaps they are implicitly assuming that the inability to provide a memory justification reflects a situation in which the “quality” of the contextual information has “degraded” to a point that it can no longer be accessed and/or verbalized. This conceptualization, however, appears to negate the basic distinction between the “accessibility” of information, and the quality (e.g., fidelity) of the information that is accessed, which (as discussed earlier) appears to provide the general rationale for the present research questions and hypotheses.

Indeed, the potential *causal* connection between the amount and quality of represented contextual content on the one hand, and the accessibility of that content, together with the “item” content, on the other, raises a critical issue regarding how one might interpret a confirmation of the hypothesized result. Consider again the general hypothesis, that the accessibility of a memory representation declines over time, whereas the content (both item and contextual) of the representations that are/can be accessed at different points in time, remains stable. If supported by the findings, they might be taken to imply that “accessibility” and “quality” (e.g., fidelity) are *dissociable* aspects of a memory representation, with one, but not the other, being affected by the passage of time. Alternatively, however, such findings might reflect a situation in which the content of some memory representations degrades over time, whereas, for various reasons, the content of other memory representations does not, with the memory representations that do not degrade (or degrade negligibly) being those that remain accessible. This possibility follows from the commonly accepted notion that successful retrieval depends on achieving a sufficiently good match between the available retrieval cues and the stored contextual information. A confirmatory pattern of results would be consistent with either of these two alternatives, yet they could hardly be more different in their broader theoretical implications. I would stop short of asking for a design that could tease these two possibilities apart[*], but I do think that the issue should be raised in the theoretical rationale that introduces and motivates the study’s research questions and hypotheses (and ultimately, in discussing the results, should they turn out as expected).

[*Logically speaking, a crucial aspect that might distinguish the two possibilities is whether the “content” that is being referred to (and examined) is content that can be used as an effective retrieval cue. As mentioned earlier, the presently proposed instructions appear to direct participants specifically toward content that can/did serve as a retrieval cue, rather than to additional content that may also ultimately constitute part of the recollective experience.]

[H5-H6] It is also hypothesized that the linguistic content of the memory justifications will remain unchanged over time, in terms of concreteness, tentativeness, and certainty [H5], and in terms of unigram (word?) frequencies [H6]. H5 appears to focus on salient qualitative aspects that might be expected to change over time, whereas the rationale for H6 is less clear. Again, as discussed above, it appears that these hypotheses should also be conditionalized on correct recall. More substantively, given that the memory “triggers” reported by the participants are being treated as (potential) memory justifications, it would be informative to evaluate not only the participants’ certainty/tentativeness (two poles of a single variable?) regarding the veracity of the triggers themselves (indirectly, via the language used to describe them), but also whether such certainty/tentativeness (as well as other qualitative and quantitative aspects of the reported contextual content) translates into certainty/tentativeness (i.e., confidence) regarding the veracity of the recalled item itself.

In other words, I think that a lot of additional valuable information (both theoretical and “practical”) would be gained by adding some measure of subjective confidence in the correctness of the recalled item. This variable could then be correlated with various measured aspects of the verbal justifications to determine which are, in fact, related to a person’s own subjective justification of the veracity of the “target” memory, and whether these relationships (as well as rated subjective confidence itself; see following comment) change over time. [Confidence ratings should perhaps be elicited after the verbal-justification response/phase, in order to minimize the potential contamination of the verbal justifications themselves.]

Finally, I think that some of the preceding issues warrant a further consideration and elaboration of H2, which, as presently formulated in the proposal, does not explicitly relate to the issue of stability or change over time (but I believe that it should).

H2 holds that items recalled with justification are more likely to be correct than items recalled without justification. This is a straightforward (“safe bet”) hypothesis that follows directly from the well established positive relationship between recollective experience and/or the recall of contextual details, and the accuracy of the recall (or recognition) response per se. What is less straightforward, however, is the extent to which this positive relationship remains stable over time. That is, is the retrieval of recollective/contextual content (i.e., the “raw materials” for memory justifications) equally diagnostic of recall accuracy after longer compared to shorter retention intervals? What should we expect and why (based on relevant previous findings and theoretical inferences, including those relating to H3-H6)? I believe that in the context of the proposed study (particularly given its initial motivation in terms of the need for, and potential value of, epistemic vigilance), this is an important question/hypothesis that needs to be directly addressed. Also, if subjective confidence ratings are added, hypotheses regarding changes in “justification” content and diagnosticity over time can and should be related to hypotheses (and relevant literature) regarding changes in subjective confidence, and its diagnosticity, over time. When this is done, the authors will face an additional challenge in reconciling their hypotheses with various studies in the metamemory literature indicating that the relationship between subjective confidence and recall accuracy does in fact weaken over time (e.g., Goldsmith, Koriat, and Pansky, 2005; Shapira and Pansky, 2019).

Methodological Comments

On the whole, the study appears to be well designed to examine the six related empirical questions presented as Hypotheses 1 to 6. My one major concern, regarding the formulation of the question used to elicit the “memory justification” responses, was discussed earlier (see comment #1, above). However the authors decide to respond to this issue, it is important that the precise wording of the question be clear, as well as any supplementary instructions or examples (to be) given to the participants.

Another potential problem, also mentioned above, concerns the time delay between the recall test and the memory justification elicitation phase. Participants are essentially being asked to “recall their earlier recall” process for each item, introducing potential 2nd-order memory and/or “narrative” contributions to stability or change in the data purported to reflect the 1st-order memory representations (and processing). While it is clear why the authors prefer not to elicit written justifications in real time following the recall of each item, perhaps they might consider using some type of “think-aloud” protocol that would only minimally disrupt the natural recall process, which could then be recorded and analyzed by methods similar to those proposed for the “offline” written verbalizations. [Perhaps both online and offline methods might be used and compared in separate experiments.]

The design includes a single “short delay” of 1.5 minutes, and two different “long delays”—1 day and 1 week (manipulated between participants). Yet, the hypotheses refer simply to “short vs. long delays.” Do the authors hypothesize that equivalent stability will be observed after 1 week compared to after 1 day?

Morris Goldsmith

Reviewer #3:

Remarks to the Author:

In this Registered Report the authors ask if written reports (“justifications”) of why specific study items were recalled change as a function of study-retrieval delay. The authors state 6 specific hypotheses to be tested, outline a plan for testing each, and describe how alternative outcomes can be assessed. Briefly, the authors have a target N = 200 participants who will study 2 lists of words, which will then be tested after an immediate period and one of 2 different delayed recalls (1 or 7 days). After recall, participants will provide written justifications for each retrieved word, which will be analyzed in several different ways.

The proposed work would be timely. The memory field continues to experiment with naturalistic approaches to research questions, and this manuscript can inform our understanding of a very “traditional” memory task—free recall of studied words—using a “naturalistic” justification procedure. In contrast, a great deal of prior research has focused on metacognitive judgements using ordinal-scale confidence ratings, feelings of knowing, etc. that do not provide nearly as rich a response from participants. The authors make excellent use of emerging options for data collection and analysis, including Prolific to facilitate data collection and natural language tools such as van Genugten & Schacter’s (2022) automated tool for justification transcript scoring. Table 1 provides a clear description of how each empirical question will be asked/assessed, and the target p-values or Bayes Factor scores are set conservatively. The proposed methods are generally well described and appear to be sufficiently powered. In my comments below, I will identify a few areas that would benefit from clarification. My other concerns center around the framing of the questions more than the proposed methods, and I will unpack these in detail as well.

1) Justifications are divided into 8 categories, with the authors citing Unsworth (2017) for category selection. Can I ask why the authors used this approach rather than using the categories from the Autobiographical Interview, which is already used to count the number of details in each description?

2) In cases, where a justification relates to more than one category, is it given a weighting of “1” for each relevant category, or is it divided in some way (e.g., does a word with 2 category justifications contribute 0.5 to each category)?

3) If more than one rater will be assigning justifications to different categories, will different raters be trained using pilot data to ensure consistency? What is the minimum interrater reliability the authors plan to achieve?

4) As initially described by Tulving (1985), autothetic consciousness encompasses personal time extending into the past as well as the future. In other words, it allows for “mental time travel” in the form of memories or episodic future thinking/episodic simulation (e.g., Szpunar et al., 2010). Given that the focus of this work is about the subjective re-experiencing of past events, I might encourage the authors to reduce some of their theoretical reliance on autothoesis and instead to more clearly invoke the reality monitoring framework proposed by Johnson and others (e.g., Johnson et al., 1988 or Johnson, 2006, which is already cited by the authors).

5) As a related point, there is evidence that the amount of available contextual information associated with a memory may serve the diagnostic function of separating events that have been personally experienced from those that have been imagined (or, presumably, heard from another source). Support for this hypothesis has typically come from ordinal-scale rating comparisons (e.g., D’Argembeau & van der Linden, 2004; Szpunar & McDermott, 2008; Arnold et al., 2011; see McDermott & Gilmore, 2015 for a review). Provided sufficient ELIs occur, a comparison of the details provided for correctly and incorrectly recalled items may allow for a direct test of predictions of this contextual information model that was not previously possible and would further ground this research in recent theory about how humans decide if a given thought reflects a “real” prior experience.

6) The authors note that a first-person perspective may help with decisions about memory accuracy (lines 37-38), but this is an unnecessary oversimplification of the literature given what I understand to be the goals of the proposed experiment. The perspective taken during recall seems to depend in part on the specifics of a provided cue, such as its age or the type of event it relates to (Rice & Rubin, 2009; 2011) and varies in general across individuals (Berg et al., 2021). The relation between vividness (which one might take as a proxy for reinstating experiences) and visual perspective is non-linear (Berg et al. 2021).

7) On Lines 58-61, the authors state that justifications have never been studied with respect to the passage of time. This may be true for complex verbal reports of the type used here, but it glosses over areas such as eyewitness testimony or metacognition that have considered this basic question using confidence ratings or other “simple” justifications for decades (e.g., Shimamura & Squire, 1988; Sauer et al., 2009).

Version 1:

Decision Letter:

Dear Mr Gamoran,

Thanks very much for revising the manuscript based on our and the reviewers' comments. Before sending this revised version for review, one of our previous editorial requests still needs to be addressed. We ask for such revisions prior to rereview in the interest of authors, to improve efficiency of the process, where it is apparent that this will diminish the chances that future additional iterations of revisions involving peer reviewers are needed.

More specifically, for the power analysis/sampling plan, you switched to a Bayesian experimental design, determining that sampling will stop once there is evidence of $BF > 10$ or $BF < 0.1$ for H3. This approach is acceptable (as explained in our previous decision letter), but requires additional information.

First, please provide a justification for why H3 is the key hypothesis for determining the final sample.

Second, ideally based on suitable simulations, please provide information on whether a sampling plan that targets H3 is likely to yield a sample that is sufficiently sensitive to also test the other hypotheses. Hypotheses for which it is likely that the sample remains underpowered (incl at the specified feasibility cut-off of 500 participants) should be listed as preregistered exploratory analyses, rather than hypotheses (in the text and the Design Table).

In sum, we invite you to revise your Stage 1 Registered Report and provide justification for the sampling plan. Please highlight all changes in the manuscript text file.

* Include a “Response to reviewers” document detailing, point-by-point, how you addressed each referee comment. If no action was taken to address a point, you must provide a compelling argument. This response will be sent back to the reviewers along with the revised manuscript.

* Ensure that you use our template for Stage 1 Registered Reports to prepare your revised manuscript:

https://www.nature.com/documents/CP_Template-RR-Stage1.docx

Failure to ensure that your revised Stage 1 submission meets our requirements as specified in the template will result in your submission being returned to you, which will delay its consideration.

* In your cover letter, please include the following information:

--An anticipated timeline for completing the study if your Stage 1 submission is accepted in principle.

--A statement confirming that you agree to share your raw data, any digital study materials, computer code, and laboratory log for all eventually published results.

--A statement confirming that, following Stage 1 in principle acceptance, you agree to register your approved protocol on the Open Science Framework (<https://osf.io/>) or other recognised repository, either publicly or under private embargo, until submission of the Stage 2 manuscript.

--A statement confirming that if you later withdraw your paper, you agree to the Journal publishing a short summary of the pre-registered study under a section Withdrawn Registrations.

Link Redacted

We hope to receive your revised manuscript within four to eight weeks. If you cannot send it within this time, please let us know. We will be happy to consider your revision so long as the report still represents a significant contribution to the literature at that stage.

* **TRANSPARENT PEER REVIEW:** Communications Psychology uses a transparent peer review system. This means that we publish the editorial decision letters including Reviewers' comments to the authors and the author rebuttal letters online as a supplementary peer review file. We publish these records for all accepted manuscripts. However, on author request, confidential information and data can be removed from the published reviewer reports and rebuttal letters prior to publication. If your manuscript has been previously reviewed at another journal, those Reviewers' comments would not form part of the published peer review file.

Communications Psychology is committed to improving transparency in authorship. As part of our efforts in this direction, we are now requesting that all authors identified as 'corresponding author' on published papers create and link their Open Researcher and Contributor Identifier (ORCID) with their account on the Manuscript Tracking System (MTS), prior to acceptance. ORCID helps the scientific community achieve unambiguous attribution of all scholarly contributions. You can create and link your ORCID from the home page of the MTS by clicking on 'Modify my Springer Nature account'. For more information please visit www.springernature.com/orcid.

Sincerely,

Hu Chuan-Peng, PhD
Editorial Board Member
Communications Psychology
orcid.org/0000-0002-7503-5131

Version 2:

Decision Letter:

Dear Mr Gamoran,

Thank you once again for your manuscript, entitled "Why we remember now and later: examining memory justifications over time," and for your patience during the peer review process.

Your manuscript has now been evaluated by the same 3 reviewers as before and they are all satisfied with your revision. We are very interested in the possibility of proceeding further with your submission in Communications Psychology, but would like to invite you to revise your manuscript so that they align with the format of our Stage registered report before we move to in principle acceptance and Stage 2 submission.

Although all required changes are presentational, we emphasize that some of these have conceptual implications that are critical Registered Reports and affect evaluation at Stage 2. More specifically, apart from minor stylistic revisions, the Introduction cannot be altered from the approved Stage 1 submission. We will therefore not be able to grant Stage 1 in-principle acceptance unless all requirements are met.

First, the presentation of the hypotheses requires significant changes. The test presently described as "Hypothesis 1" is not a hypothesis, it's a validation check (as also indicated in the Design table). This means that the analysis must be listed as a paradigm validation check only, with a clear statement that if the expected results of this analysis (delay-dependent forgetting effect) are not confirmed, the data are uninterpretable, thus failing the requirements for Stage 2 Registered Reports. Although this outcome is exceedingly unlikely (especially given the piloting work), this separation of baseline

checks and hypotheses and the implications for further consideration are critical. This analysis should not be listed in the Design table.

Second, your Hypothesis 2 currently includes 2 hypotheses (2 predictions) that pertain to the same research question. These must be listed separately in the main text and in the design table (new numbering H1 and H2) and you must include a statement as to how you would interpret a positive outcome for one hypotheses (e.g. the main effect) and a negative or inconclusive finding for another (e.g. the interaction).

Apart from these issues, the Design Table contains nearly all necessary information, but some aspects are reported in the wrong place, or redundantly. Please refer to the attached documents for some guidance on how information should be distributed. Please note that you must carefully review the information for all 3 hypotheses and all exploratory analyses, to ensure it is complete and stated as intended.

The entire "Appendix C" belongs in the main protocol, together with the power analysis/sampling plan. This information cannot be relegated to the Supplement but should appear in the context of the statement: "To assess the effects of delay duration on level of detail (Hypothesis 3), Bayesian t tests will be conducted. The sampling plan for these tests is the collection of a sample size required for a Bayes Factor score of 10 in support of the null hypothesis, or a comparable size 367 in support of the differing hypothesis, or until reaching the constraint of N = 500 participants"

Please refrain from calling the planned exploratory analyses "hypotheses" (this is still the case in the main text).

Please remove the limitations statement. This does not form part of a Registered Report stage 1 protocol. Limitations will need to be discussed at length at Stage 2, in the Discussion section.

On a purely stylistic note, the "Introduction" section should not include subheadings, the "Methods" should be called "Analysis plan" and the "Appendix" should be renamed "Supplementary Information".

* Ensure that you use our template for Stage 1 Registered Reports to prepare your revised manuscript:

https://www.nature.com/documents/CP_Template-RR-Stage1.docx

Failure to ensure that your revised Stage 1 submission meets our requirements as specified in the template will result in your submission being returned to you, which will delay its consideration.

* In your cover letter, please include the following information:

--An anticipated timeline for completing the study if your Stage 1 submission is accepted in principle.

--A statement confirming that you agree to share your raw data, any digital study materials, computer code, and laboratory log for all eventually published results.

--A statement confirming that, following Stage 1 in principle acceptance, you agree to register your approved protocol on the Open Science Framework (<https://osf.io/>) or other recognised repository, either publicly or under private embargo, until submission of the Stage 2 manuscript.

--A statement confirming that if you later withdraw your paper, you agree to the Journal publishing a short summary of the pre-registered study under a section Withdrawn Registrations.

Link Redacted

We hope to receive your revised manuscript within four to eight weeks. If you cannot send it within this time, please let us know. We will be happy to consider your revision so long as the report still represents a significant contribution to the literature at that stage.

* **TRANSPARENT PEER REVIEW:** Communications Psychology uses a transparent peer review system. This means that we publish the editorial decision letters including Reviewers' comments to the authors and the author rebuttal letters online as a supplementary peer review file. We publish these records for all accepted manuscripts. However, on author request, confidential information and data can be removed from the published reviewer reports and rebuttal letters prior to publication. If your manuscript has been previously reviewed at another journal, those Reviewers' comments would not form part of the published peer review file.

Communications Psychology is committed to improving transparency in authorship. As part of our efforts in this direction, we are now requesting that all authors identified as 'corresponding author' on published papers create and link their Open Researcher and Contributor Identifier (ORCID) with their account on the Manuscript Tracking System (MTS), prior to

acceptance. ORCID helps the scientific community achieve unambiguous attribution of all scholarly contributions. You can create and link your ORCID from the home page of the MTS by clicking on 'Modify my Springer Nature account'. For more information please visit www.springernature.com/orcid.

Sincerely,

Hu Chuan-Peng, PhD
Editorial Board Member
Communications Psychology
orcid.org/0000-0002-7503-5131

REVIEWERS' COMMENTS:

Reviewer #1:

Remarks to the Author:

The contributors have responded thoroughly to my concerns. I am happy to recommend this stage 1 RR.

I always sign my reviews,

Jason M Chin

Reviewer #2:

Remarks to the Author:

This is a revised version of a proposed registered report that I reviewed earlier. In this revision (and accompanying rebuttal letter) the authors have satisfactorily addressed all of my previously expressed concerns. I wish the authors good luck with the continuation of the study.

Reviewer #3:

Remarks to the Author:

The authors have thoughtfully updated their manuscript in response to the prior feedback. I have no new comments, but do want to make sure that their new plan is to use Internal and External detail counts only from the automated autobiographical interview scoring approach they now describe. If they do, then I don't think additional changes are required. If they still wish to use detail category-level information, as they described in their initial submission, then that should be clarified going forward.

Version 3:

Decision Letter:

23rd Jan 2025

Dear Ms Raz Groman,

Thank you once again for submitting your revised Stage 1 Registered Report, entitled "Why we remember now and later: examining memory justifications over time." Everything is in order and I am delighted to say that we can offer acceptance in principle. You may progress to Stage 2 and complete the study as approved.

As you know, a condition of in-principle-acceptance is that the authors agree to deposit their Stage 1 accepted protocol in a repository, either publicly or under embargo until Stage 2 acceptance and publication. We are very keen to showcase our in-principle accepted protocols, so that our readers, reviewers, and potential authors can gain insight into the requirements of the format as well as an idea of the types of projects that are suitable for publication in Communications Psychology. We have set up a space on figshare to host all of our in-principle accepted protocols, which can either be made public or kept under embargo until Stage 2 acceptance (depending on author preference). This gives you the opportunity to have your work publicly associated with Communications Psychology, and of course we will be very pleased to showcase your report if you agree to share it publicly.

Depositing the work on our figshare space does not preclude deposition of your Stage 1 protocol on other depositories – your protocol can also be posted on OSF, Dataverse, Dryad or any other public repository of your choice. You also do not

need to do anything – if you agree with posting your protocol on our figshare space, we will upload your protocol on your behalf and either set it public or place it under embargo, depending on your choice. Your protocol will be licensed under a CC BY license (Creative Commons Attribution 4.0 International License). The CC BY license allows for maximum dissemination and re-use of open access materials and is preferred by many research funding bodies. Under this license users are free to share (copy, distribute and transmit) and remix (adapt) the contribution including for commercial purposes, providing they attribute the contribution in the manner specified by the author or licensor (read full legal code: <http://creativecommons.org/licenses/by/4.0/legalcode>) Please note that any use of <https://springernature.figshare.com> will be subject to the Figshare terms of use. Figshare has the right to enforce these terms and conditions where applicable. Use of third party services and sites will be subject to the relevant terms of use and will apply if we act on your behalf in this regard. Do let me know if you would like to take up this option or if you have any questions regarding the protocol deposition requirement.

IMPORTANT:

In cases where the registered experimental design is altered after AIP due to unforeseen circumstances (e.g. change of equipment or unanticipated technical error), the authors should consult the editors immediately for advice, prior to the completion of data collection.

Following completion of your study, we invite you to resubmit your paper for peer review as a Stage 2 Registered Report. Please note that your manuscript can still be rejected for publication at Stage 2 if the Editors consider any of the following to hold:

- The results were unable to test the authors' proposed hypotheses by failing to meet the approved outcome-neutral criteria
- The authors altered the Introduction, rationale, or hypotheses, as approved in the Stage 1 submission
- The authors failed to adhere closely to the registered experimental procedures without previously seeking editorial approval
- Any post hoc (unregistered) analyses were either unjustified, insufficiently caveated, or overly dominant in shaping the authors' conclusions
- The authors' conclusions were not justified given the data obtained

We encourage you to read the complete guidelines for authors concerning Stage 2 submissions at <https://www.nature.com/commpsychol/submit/registered-reports> and <https://www.nature.com/documents/commpsychol-style-formatting-checklist-article-rr.pdf>.

Please especially note the requirements for protocol deposition, data sharing, and that withdrawing your manuscript will result in publication of a Retracted Registration.

When you are ready, please use the following link to access your home page and submit your Stage 2 Registered Report:

Link Redacted

*This url links to your confidential homepage and associated information about manuscripts you may have submitted or be reviewing for us. If you wish to forward this e-mail to co-authors, please delete this link to your homepage first.

* **TRANSPARENT PEER REVIEW:** Communications Psychology uses a transparent peer review system. This means that we publish the editorial decision letters including Reviewers' comments to the authors and the author rebuttal letters online as a supplementary peer review file. This means that the records will be published together with your Stage 2 report. On author request, confidential information and data can be removed from the published reviewer reports and rebuttal letters prior to publication. If your manuscript has been previously reviewed at another journal, those Reviewers' comments would not form part of the published peer review file.

We expect your Stage 2 Registered Report to be submitted by the date specified in your latest cover letter. If unforeseen circumstances prevent submission by that date, please contact us as soon as possible to discuss any changes to the submission time-frame.

Thank you again for offering us this work and we look forward to receiving your Stage 2 Registered Report.

Yours sincerely,
Marika Schiffer, on behalf of

Hu Chuan-Peng, PhD
Editorial Board Member
Communications Psychology
orcid.org/0000-0002-7503-5131

Version 4:

Decision Letter:

9th Oct 2025

Dear Ms Raz Groman,

Thank you once again for submitting your Stage 2 Registered Report, entitled "Why we remember now and later: examining memory justifications over time," and for your patience during the re-review process.

Your manuscript has now been evaluated by two of the original reviewers, whose detailed comments are included at the end of this letter. In the light of our reviewers' advice, we are pleased to inform you that we will be able to accept your Stage 2 manuscript, pending revisions to address some reviewers' comments and editorial requests.

To guide the scope of the revisions, the editors discuss the referee reports in detail within the team, including with the chief editor, with a view to (1) identifying key priorities that should be addressed in revision and (2) overruling referee requests that are beyond the scope of Stage 2 Registered Reports.

The reviewers (especially Reviewer #2) highlight that the Stage 2 Report (as the Stage 1 Report previously) does not fully engage with the main effect of time. The reported effect in Table 3 appears significant (memory declines over time for both conditions). This should be suitably reflected in the Discussion.

Both reviewers provided feedback on some arguments brought forward in the Discussion section. Please respond to these concerns through appropriate revisions. Please note that the Introduction should not be changed.

The manuscript is overall highly compliant with the journal's requirements for statistics reporting, but a few issues need to be addressed. The referees also call for greater detail in the stats reporting and we ask you to respond to this request in line with journal guidelines. We do not consider further analyses of the data necessary, but if you want to add exploratory analyses as per the referee's suggestion, you are welcome to do so.

One of the main reasons for delays in eventual acceptance is failure to fully comply with editorial policies and formatting requirements. To assist you with finalizing your manuscript for publication, I attach our Editorial Requests Table which lists all of our editorial policies and formatting requirements.

Please attend to *every item* in the Table and upload a copy of the completed checklist with your submission.

OPEN ACCESS:

Communications Psychology is a fully open access journal. Articles are made freely accessible on publication. For further information about article processing charges, open access funding, and advice and support from Nature Research, please visit <https://www.nature.com/commpsychol/open-access>

* **CODE AVAILABILITY:** To proceed to formal acceptance, you must now publicly deposit the custom analysis code supporting your conclusions; please use a repository that mints the code with a digital object identifier (DOI). The manuscript must include a section titled "Code Availability" at the end of the methods section. The link to the repository and the DOI must be included in the Code Availability statement. Publication as Supplementary Information will not suffice.

* **DATA AVAILABILITY:**

It is a requirement for Registered Reports to make data publicly available. All Communications Psychology manuscripts must include a section titled "Data Availability" at the end of the Methods section. More information on this policy, is available in the Editorial Requests Table and at <http://www.nature.com/authors/policies/data/data-availability-statements-data-citations.pdf>. Please share a link to your publicly deposited data in the Data Availability statement.

We hope to hear from you within [ENTER TIME PERIOD]; please let us know if the revision process is likely to take longer.

Please use the following link for uploading the materials:
Link Redacted

With best regards,
Marika, on behalf of

Hu Chuan-Peng

Hu Chuan-Peng, PhD
Editorial Board Member
Communications Psychology
orcid.org/0000-0002-7503-5131

Marika Schiffer, PhD
Chief Editor
Communications Psychology

Reviewer #2:

Remarks to the Author:

This is an excellent study on a timely topic, built upon and extending an increasingly influential proposal on the nature of forgetting. As fitting a stage-2 registered report, the authors have done what they set out to do in the approved stage-1 proposal and obtained an interesting pattern of results. The article certainly deserves to be published. Nevertheless, there are still some issues in the present manuscript that I think should be addressed before publication.

Main Comments

1. The main question asked in the present study concerns whether the fidelity (content and validity) of memory justifications remains stable over time, and if so, what can we infer regarding the responsible memory mechanisms. On pages 4-5 of the current manuscript, the authors put forward the theoretical implications that might be drawn regarding the underlying memory mechanisms under two alternative outcomes, with decreased fidelity over time most naturally interpreted as reflecting a “graded decaying” of the underlying contextual information, and stable fidelity most naturally interpreted as reflecting a “dissociation of accessibility and fidelity” – by which the accessibility of memory traces decreases over time while the contents of those traces (including contextual information) remain stable. After doing so, however, the authors go on to acknowledge (top of p. 5) that the theoretical implications of a finding of stable justification fidelity over time are not completely straightforward, as there is an alternative explanation by which the same pattern might occur precisely because accessibility and fidelity are tightly linked. After setting out this alternative explanation (which invokes graded rather than all-or-none forgetting of memory content), the authors acknowledge that if the findings indicate similar levels of justification fidelity over time, “future studies will be needed to elucidate between these two theoretical accounts.”

So, this is just a “gentle reminder” that the existence of this alternative explanation, with the implied need for caution in reaching theoretical conclusions, should be returned to in Discussing the Results, lest the reader forget that there is in fact a different interpretation than the one promoted by the authors. (Of course, if the authors can use some of the more detailed results to argue against the competing explanation, they should do so.)

2. Regarding the specific question of whether the validity of memory justifications remains stable over time, there appears to be one result, not addressed by the authors, that counts squarely against this conclusion. This result relates to an ambiguity in the analysis plan that was not noticed at stage 1, but I don't think that it can simply be ignored for that reason. In the approved stage-1 proposal, the authors formulated Hypothesis 2 (which in the present manuscript is split between Hypotheses 1 and 2) as: “Do justifications reflect mnemonic performance?”, and then elaborated: “Higher recall accuracy will be found for items recalled with a justification than items without a justification, supporting the claim that justifications reflect mnemonic performance. NO SIGNIFICANT INTERACTION will be found with TIME DELAYS, such that ACCURACY FOR ITEMS WITH JUSTIFICATIONS WILL NOT DECREASE OVER TIME, further supporting the validity of justifications.” (The second part of this elaboration, emphasized IN CAPS, is essentially the same in the current manuscript.)

The problem with this formulation is that the absence of a Time x Justification interaction on recall accuracy and the absence of a simple effect (decrease) of Time on the accuracy of justified items are two different ways of looking at the relationship between memory justification and accuracy over time that can and do (in the present case) yield divergent conclusions. In line with their formal analysis plan, in the present manuscript the authors based their conclusion entirely on the absence of a Time x Justification interaction, concluding that because the difference in the accuracy of justified versus non-justified responses remained equally large (statistically), regardless of delay: “justifications remain valid indicators of retrieval accuracy, even after a delay.” This is true; but this does NOT say that the degree of validity (what can be inferred regarding the likely accuracy of the response, given that it was justified) remains stable over time. This latter aspect is what the authors

“promised” to check in the Introduction to the present manuscript (p. 6):

“With regard to effects of time, we further predicted that the accuracy of items with justifications will not decrease over time, contrary to what would be predicted if justifications of true memories are prone to increased confabulation over time, in which case the proportion of items with justifications which are correct recalls ought to decrease (Hypothesis 2).”

However, the authors did not analyze (report) the corresponding simple effect. Examination of Figure 2 (p. 27) indicates that the accuracy of responses with justifications did in fact decrease significantly (based on the error bars) over time, from about 98% (the exact mean is not given) after the short delay, to around 79% after the longer delay. Cast in terms of output-bound accuracy (conditional probability correct), when a participant in this study provided a justification for a recall response after a short delay, that answer could be strongly relied upon, with 0.98 probability of being correct. After the longer delay, however, a justified recall response could only be “modestly” relied upon, with an 0.79 probability of being correct. Accordingly, if we hold the authors to their earlier statement quoted above, there does indeed appear to be evidence that justifications are prone to “increased confabulation” over time.

The upshot of all of this is that to fully and fairly address the questions they have raised (and hypotheses they have put forward), I believe that the authors need to integrate this finding into the article and adjust their conclusions accordingly.

Specific comments:

1. Abstract, second sentence. I suggest to replace this sentence with one that explains what is meant by “Memory Justifications.”
2. Abstract, line 16, sentence beginning “These finding indicate...” I think that for most readers, the evidential basis for this conclusion will be unclear. Also, given the new evidence of decreased validity after a delay (cf. comment #2 above) the formulation of what the results indicate should be tweaked accordingly.
3. Abstract, end. Consider “softening” the concluding sentence.
4. p. 5, line 119. The term “triggered” appears to be a left-over from the pilot instructions.
5. p. 16, line 351. Find a way to indicate that confidence ratings were provided for all recalled items, including those that were not justified.
6. p. 32, Non-pre-registered exploratory analyses. I suggest to add the correlation between amount of detail and confidence, and to provide a full correlation table. I also suggest to report a two-way analysis, Time x Justification, on the confidence ratings, instead of the present t-test.
7. p. 33, lines 584-585. Instead of “on memory of justifications,” perhaps: “on the contextual memories that constitute memory justifications.”
8. p. 33, first full paragraph. The second part of this paragraph, concerning the stability of “the relationship between memory justification and accuracy” (which is an overly vague way to formulate the specific aspect of interest; see main comment #2) should be revised in light of the new evidence of significantly decreased accuracy of justified recall responses over time.
9. p. 34, lines 615-618. The authors wrote: “Since no contextual cues were explicitly manipulated or removed between encoding and retrieval, the observed forgetting is more consistent with a failure to internally reinstate the appropriate retrieval context, rather than with the selective degradation of contextual features themselves.” I am unable to understand what the authors mean by this sentence (i.e., the logical connection between the two parts), and for that reason, am also unsure as to whether this sentence was intended to relate to the alternative mechanistic explanation (main comment #1), by which, after a delay, the degradation of stored contextual representations for some/many of the items might reduce the potential match between those representations and the reinstated retrieval cues, thereby limiting successful recall to those items whose contextual representations have remained relatively intact. Such a possibility does not in any way diminish the importance of context reinstatement (internal or external; primarily internal) as a key memory “access” mechanism. Quite the contrary. It merely assumes that “effective” reinstatement (successful access by way of reinstated cues) become less likely to the extent that the internally/externally reinstated contextual information is no longer represented in memory with sufficient fidelity to achieve a good “match.”
10. p. 34, line 625. In line with the preceding comment, I think the word “compelling” is a bit too strong.
11. p. 35, line 647. It is of course true that word concreteness can be influenced by the pragmatics of language use. However, the simplest interpretation would be that the accessible justification information became less concrete over time.
12. p 35, lines 650-653. Is it valid to infer “low” confidence from the absence (or low rate of) “certainty expressions”? My guess is that you can infer high confidence from the presence of “certainty expressions” and low confidence from the presence of “tentativeness expressions,” but much less (if anything) from the absence of either of these types of expressions. In any case, I think it might be more informative to compare the 2-way pattern, Time x Justifications, on confidence versus accuracy (and see next comment).

13. p. 36, lines 654-656. It is true that the proportion of justified items remained stable over time. However, the proportion of justified items that were correct, did not. This latter decline (from 0.98 to 0.79) presumably corresponds quite well to the decline in confidence ratings over time.

14, p. 36, lines 662-664. The idea that linguistic aspects of justifications might be predictive of response accuracy, above and beyond that which can be achieved by explicit confidence ratings, is worth examining. Perhaps a suitable analysis can be added as an additional exploratory analysis?

Morris Goldsmith

Reviewer #3:

Remarks to the Author:

This is a follow-up submission to a preregistration report I reviewed some months ago. In this work, the authors focus on written justifications following successful retrieval and ask if the presence and/or contents of justifications change across short or long retrieval delays. Online participants studied lists of words and were tested either after 1.5 min or 1 day delays via a free recall format. All retrieved words were then presented as cues to elicit written justifications. As expected, longer delays were associated with fewer recalled items. Recall accuracy was higher for items accompanied by justifications. No differences were observed between delays in the proportion of episodic details included in justifications. Across several additional pre-registered analyses, it was found that the proportion of items with a justification did not differ across conditions, "concreteness" had a mixed pattern across delays, "tentativeness" increased with a longer delay, and "certainty" seems no to differ. Word usage did not differ across delays. Confidence appeared to track some aspects of justification content and overall decreased with a longer delay. The authors conclude that verbal memory justifications are "robust" indicators of recall accuracy (p. 38).

Consistent with remarks from my initial review, this is a timely experiment that seems to offer an excellent mix of traditional laboratory experimental control and a more "naturalistic" approach to understanding an everyday process related to memory. There is a useful mix of traditional null hypothesis and Bayesian testing that allows for null results to be better interpreted, while also offering some cautionary notes on some "weakly" significant results. The authors were able to provide clear answers to each of their questions/hypotheses. In short, my general enthusiasm for this work has not diminished since my initial review. My concerns fall primarily on conceptual factors that would benefit from clarification, and I will discuss these in more detail below.

1. The authors state on line 51-53 that recollection "is an adaptation meant to... [distinguish] between veridical and non-veridical memories. Mahr and Csibra (2017, cited by the authors) certainly make this strong claim, but this seems less consistent with the general position taken by Schacter, Tulving, Suddendorf, and their colleagues in the other citations associated with this claim (refs 32-34), which instead focus on the centrality of recollection/episodic details in the imagination/simulation of future events. Were the latter 3 references intended to relate to this particular assertion? If so, could the authors be more explicit in how the two views (ref 8 vs 32-34) are mutually supportive, rather than distinct from one another?

2. As a future direction (lines 696-700), the authors suggest using materials that are more complex (eyewitness accounts or narrative descriptions of film clips). This is a logical extension that would continue to mix laboratory control and naturalistic generalization. Would the authors be willing to expand their thinking on this suggestion in more detail? For instance, would they expect that justifications of a single word from a list would perhaps approximate those of a single detail from a more complex narrative, meaning that some aspects would be remembered (most of which would have accompanying justifications) while others will be forgotten? The alternative expectation would seem to be that the whole of an event would be (generally) recallable if it was accompanied by a justification. These are obviously divergent predictions, and it would be helpful if the authors could make clear their expected link between a word list and eyewitness event description.

3. With respect to the (often null) effects of delay on justification w, the authors noted on lines 645-647 that time seems to impact "pragmatic" language use but not the amount of associated detail. A fairly lengthy paragraph precedes this conclusion, in which the authors review the various results on concreteness and other word scores. However, given the correlation between delay and confidence, and between confidence and tentativeness/concreteness/certainty (lines 567-577), might the most straightforward explanation be that time, per se, has a minimal impact on language use, and that it is instead confidence that is the main driver of the observed effects of delay? This idea is partially engaged with in the following paragraph, but explicit statement one way or the other would likely allow for a tighter overall discussion of this portion of results.

REVIEWERS' COMMENTS:

Reviewer #1:

1. Abstract – I'm not sure the word "autonoetic consciousness" should be in the abstract. Many will not know what this means and its meaning is not made clear until one starts reading the Introduction. And, I'm not sure it adds much to the Abstract.

Response: The abstract was revised and the term removed (p. 1)

2. Abstract – Could it say "free recall performance" instead of "objective free recall performance". I'm not sure what objective means in this context.

Response: The abstract was revised accordingly (p. 1)

3. Will the study materials be made available (e.g., the word lists)?

Response: Yes. The data availability statement was revised to state explicitly that all materials (including word lists) will be made publicly available on the OSF website (p. 23): "The pilot study data is freely available at the Open Science Framework (OSF) website (<https://osf.io/3wu5a/>). All study materials (e.g., word lists) and all behavioral data collected via the online study will be anonymized and made freely available at the same location."

4. I am not sure the study is reproducible. For instance, many methods are described in a high level of abstraction (see the Justifications section). What will participants be told exactly? The protocol refers to the "Autobiographical Interview" and its scoring tool. Many readers will not know what these are and what they entail.

Response: The Methods section was revised for clarity and specification, including addition of the wording of instructions given to participants (p. 14; Instructions are also made publicly available along with study materials): "Participants' introspective reflections regarding the information that triggered recall, or "justifications", will be self-reported. Instructions before the justification-recall phase are: "You will now be shown again the words you just remembered. Please explain in as much detail as possible why you think this particular word appeared in the study phase. Consider any factors that you think might justify your recollection of the word". During the justification-recall phase participants are shown one word at a time produced by the participant during the item-recall phase and probed with the instructions: "Please describe in as much detail as possible why you believe the word above appeared in the study phase"."

The section about the Autobiographical Interview has been further elaborated (p. 14). References have been included for the manual Autobiographical Interview^{1,2}, and for the Automated Autobiographical Interview tool used in the current study

³, as well as linking to the Automated tool (Google Colab notebook) wherein similar analyses can be run (p. 15):

“To further explore the effects of time on justifications, each justification will be analyzed for its number of internal/external details. This analysis is based on the Autobiographical Interview ^{1,2}, in which the number of details providing distinct pieces of information are counted, and then classified as internal or external to the event described. The analysis will be performed using the Automated Autobiographical Interview Scoring tool ³, a Neural Network based model for automated classification of internal and external details in a text (A Google Colab Notebook and instructions for the automated scoring can be found at: <https://github.com/rubenvangenugten/>).”

5. What are “ecological delays”?

Response: The justification of the delays chosen has been more clearly stated (p. 16), citing supporting references.:

“The one-day interval is aimed at examining an ecological delay--namely, one which corresponds to delays at which there is a considerable decrease in everyday-occurrences recalled ⁴. It has recently been argued that memory for events after delays of such timescales (12 hours -7 days) is a specific stage of Long Term Memory (“Transitional Long Term Memory” ^{5,6}). Recall at such a delay has also been tested in Pilot study B (See Appendix B).”

In addition, the additional pilot studies following changes administered to the study paradigm demonstrate the feasibility of free recall after one day, but that there is an apparent floor effect for recall (at least for lists consisting of 16-items as in the current study) following a delay of one week (see Appendix B; pp. B13-B14).

6. Power analysis – the study is powered to find the effect in the pilot study. Why? What is the justification for that?

Response: In accordance with the Registered Report guidelines for committing to a singular sample size fit for the main hypothesis, the sample size has been revised (pp. 18-19) to sampling according to a Bayesian sampling scheme until achieving $BF_{10} > 10$ (or a limit of 500 participants) for Hypothesis III (See also response to section 9):

“To assess the predicted null effects of level of detail (Hypothesis 3), Bayesian t-tests will be conducted. The sampling plan for these tests is the collection of a sample size required for a Bayes Factor score of 10 in support of the null hypothesis, or a comparable size in support of the differing hypothesis, or until reaching the constraint of $N = 500$ participants.”

7. Exclusion criteria – excluding answers that are “completely inappropriate” is subjective. What will be the safeguards so that this is not affected by the data?

Response: Exclusion criteria have been revised and stated according to objective measures (p. 18):

“The answers will be screened manually, and responses containing only nonsense words or characters and no coherent words, or containing only text irrelevant to the experiment (e.g., text

copied from a Wikipedia page) will lead to participant exclusion. Justifications will be screened for exclusion (as well as classification) by two separate raters. To ensure consistency in exclusion criteria, the Inter-rater reliability will be assessed and a standard of Cohen's Kappa of at least 0.8 will be required."

8. Hypothesis 2 – I would like the authors to more clearly spell out their decision rule in the table of hypotheses.

Response: The conclusions in the Table have been revised to state more plainly the interpretation of result outcomes (pp. 19-22), for example (Hypothesis I):

"A $BF_{10} > 10$ will be interpreted as supporting delay dependent forgetting, such that fewer items are recalled following the long delay than the short delay.

A $BF_{10} < 0.1$ ($BF_{01} > 10$) will be interpreted as providing evidence against delay dependent forgetting. $0.1 < BF_{10} < 10$, will be interpreted as inconclusive results."

9. Hypotheses using a Bayes factor of 10 as the cutoff. What will the authors conclude if the is < 10 ? I would like to see them include in the table simple conditional statements, e.g., if the $BF > 10$, we will conclude XXX and if the $BF < 10$, we will conclude XXX. Can anything be falsified in this study?

Response: Conditional statements are included in Table I (pp. 19-22). Including cases of falsification of predicted hypotheses and inconclusive findings. $BF_{10} < 10$ will be considered as inconclusive findings. $BF_{01} > 10$ will be taken as evidence for falsification. The cutoff of $BF > 10$ is according to the Nature guidelines for registered reports.

10. The Stage 1 report should include a strong constraints on generality statement. There seem to be some severe constraints on the generality of this study in terms of both the contexts it may generalize to and the people. These should be forcefully noted.

Response: Additional constraints on the generality of contexts have been included (p. 2):

"Statement of limitations

A limitation of this study is in generalizability beyond the study sample, since participants are recruited online and speak English. Hence the potential effects of cultural background on the results are yet unknown. Still, in contrast to the proposed study, pilot results of the first pilot study were not conducted in English, nor with WEIRD participants, and were not conducted online (but in person), suggesting the effects of cultural background to be limited. These factors ought to be studied in the future with a non-English-speaking, non-Western sample. This study may also be limited in differing contexts to which it may apply, this study is conducted for neutral valence items, consisting of single words per item. The ability to generalize from justification of single words to complex events such as in eyewitness testimony may be limited without further examination in such contexts."

The two pilot studies cover a variety of participants, including different cultural contexts, different age groups and different socio-economic groups, thus reducing somewhat constraints on the generality of findings.

I always sign my reviews,
Jason M. Chin (ORCID: 0000-0002-6573-2670)

Reviewer #2:

General Comments

1. "Memory justification"

The term is initially introduced in relation to the need for epistemic vigilance, to provide sufficient justification for believing that the information that one recalls (or recognizes) from memory is in fact veridical. According to the authors, such justification can be provided by (is one of the primary functions served by) the auto-noetic/recollective content/experience that typically accompanies memory recall (and, perhaps to a lesser extent, recognition). Operationally, however, memory justifications are elicited as participants' answers to the question "What *triggered* your recall of the item X?" (or equivalently, according to the flow diagram in Figure 1, "How did you remember the word X?"). There are two salient problems with this operationalization: (1) It does not ask, explicitly for a "justification" of why the participants believe that the recalled item is correct/veridical; rather, it asks for an introspection regarding what led to the emergence of the item into consciousness. This is assuming that the participants pay careful attention to the wording and clearly understand what is being asked. (2) As a consequence, it does not ask participants to report contextual recollective content/experience that does not serve to "trigger" the memory, but that may come to mind *after* the target item has been accessed. Such content is no less valuable in evaluating/justifying the veracity of the recall than is the content that proximally triggered the memory.

Response: We thank the reviewer for raising this concern. The phrasing of the instructions to participants have been carefully rephrased as part of the revision (p. 14), and a pilot study has been run with the new phrasing and design changes, (see appendix B pp. B3-B5 and further comments below). The instructions now elaborate before the justification-recalling phase:

““You will now be shown again the words you just remembered. Please explain in as much detail as possible why you think this particular word appeared in the study phase. Consider any factors that you think might justify your recollection of the word”.”

and preceding each specific justification is the shorter prompt:

““Please describe in as much detail as possible why you believe the word above appeared in the study phase”.”

Thus the new instructions explicitly ask for justification, and explaining the perceived veracity of the item (“why you think this particular word appeared...”, and in individual prompts: “why you believe the word above appeared in the

study phase”), and also point directly to the experience of recollection (“...factors...justify your recollection of this word”).

(3) Particularly when the justifications are elicited in a separate phase after all items have been recalled, it seems unreasonable to believe that participants are capable of distinguishing between contextual information that may have triggered retrieval of the target, and contextual information that may have been retrieved after (via) retrieval of the target.

Response: to address this concern, our new pilot study included two conditions for the order of justification-recollection (Appendix B, pp. B3 - B5), in one condition the justification recalling phase is a separate phase after the item recall phase (as in the first pilot and originally proposed design), the second condition included a request for justification as part of the item recall phase, such that after each item was recalled, participants were asked to recall that item’s justification. The recall-order conditions did not have a significant effect on number of items recalled, numeric confidence scores or markers of justification content, with the exception of the length of justifications (number of words per justification), which were found to be longer in the separate-justifications-phase condition. Hence the proposed paradigm for the main study remains with the separate justification-eliciting phase.

One might perhaps argue that participants’ inattention to the precise wording of the instructions and/or likely inability to recall the precise temporal order of the information that came to mind might in fact negate the basic concern that they are not providing all of the contextual information relevant to memory justification. At best, however, this is an unwanted source of inter-individual variance, and it would surely be preferable to reformulate the instructions so that they explicitly ask for the information that most closely corresponds to a true memory “justification.” This might be done by explicitly asking them to justify their responses (if possible) with relevant contextual memories, or alternatively, by asking them to report all of the contextual details that they can remember. The first method is of course more direct, but has the potential shortcoming of confounding the amount and quality of accessible information with the amount and quality of information that participants believe is relevant for memory justification. The latter approach avoids (as much as possible) the potential effects of such “editing,” but then the dependent variable would need to be relabeled (and conceived) as something like “verbally expressible recollective-contextual content that might potentially provide the basis for memory justification” (quite awkward), rather than memory justification per se.

Response: our rephrased instructions (detailed in previous response) attempt to address these concerns, by explicitly directing towards available contextual information which is used for the sake of memory justification.

We also rephrased in the manuscript's Introduction (pp. 3-5) to better clarify that this study is aimed at this type of information in use of memory justification, a subset of all forms of information which may be in use to justify memories:

“The ability to reinstate an experience along with accompanying contextual information (internal or external) may serve as an authentication device that helps us determine that we are remembering correctly ⁷...

Intuitively, one may reason that contextual information upon which justifications rely, like other information stored in memory, gradually fragment over time, losing more details and clarity as time goes by ⁸⁻¹⁰...

In the current study we ask whether justifications lose fidelity over time, in line with findings of subjective confidence assessments, or whether they remain intact in their degree of vividness, detail, and their content after a long time-delay. A *decrease* in fidelity, vividness and detail over time would be expected due to a gradual decaying of underlying contextual information, whereas *maintained* levels of fidelity, vividness and detail would be in line with a dissociation of accessibility from fidelity. In the latter case, despite reduced accessibility to contextual information, that information which is still accessible will remain unchanged in degree of fidelity (demonstrating an “All-or-None” form of forgetting).”

2. Theoretical bases for the empirical hypotheses

The general theoretical rationale for the main experimental hypotheses was raised very briefly at the bottom of page 3, pointing (through numbered references) to findings indicating that under some conditions, and for certain types of memory content, the accessibility of the memory representations may decline over time, whereas the fidelity of the representations that remain accessible, is stable. After presenting this idea, the general research question was then formulated as “whether justifications lose fidelity over time or whether they remain quantitatively and qualitatively intact after a long time-delay.” Following up on my earlier concerns about the theoretical and operational definition of “memory justification,” let me just note that all of the various formulations suggesting that memory “justifications” are “stored” or “represented” in memory should be avoided. One may ask whether the memory justifications that people provide lose or retain fidelity over time, but it is the contextual content that provides the raw materials for such justifications, not the justifications themselves, that is (perhaps) stored and represented in memory, accessed, lost or degraded over time, and so forth. So, to paraphrase, the main question appears to be that given that a previously encoded memory can be accessed at a particular point in time, will all of the associated contextual content, encoded as part of that memory (that could potentially be used to provide a memory justification) remain intact/unchanged, regardless of the retention interval after which the memory is accessed?

Response: We thank you for raising these important points. We rephrased the above-quoted part of the manuscript (pp. 4-5), to better indicate that the contextual information is retrieved from memory and informs justifications:

”Intuitively, one may reason that contextual information upon which justifications rely, like other information stored in memory, gradually fragment over time, losing more details and clarity as time goes by ⁸⁻¹⁰...

In the current study we ask whether justifications lose fidelity over time, in line with findings of

subjective confidence assessments, or whether they remain intact in their degree of vividness, detail, and their content after a long time-delay. A *decrease* in fidelity, vividness and detail over time would be expected due to a gradual decaying of underlying contextual information, whereas *maintained* levels of fidelity, vividness and detail would be in line with a dissociation of accessibility from fidelity. In the latter case, despite reduced accessibility to contextual information, that information which is still accessible will remain unchanged in degree of fidelity (demonstrating an “All-or-None” form of forgetting).”

In other words, Justifications may be affected by time in one of two different ways, reflecting two different accounts of forgetting of contextual information: Decay over time would present weaker memory traces upon which Justifications rely. All or none forgetting, on the other hand, may lead to fewer traces overall, but traces which remain stay strong, leading to similarly confident and vivid justification descriptions.

As a “working hypothesis,” the authors hypothesize that the answer is “yes” — that such content and (therefore) the memory justifications that might be verbalized on the basis of that content, will remain stable over time. I use the term “working hypothesis,” because beyond the first two specific hypotheses (H1 and H2), which are rather trivial (clearly supported by “common sense” and/or existing results), the key hypothesis of “stability” in the remaining four hypotheses (H3-H6) is justified solely on the basis of the corresponding observed trends in the pilot data. No specific theoretical justifications are provided. This is problematic for several reasons. First, the conditions of the pilot study are substantially different from those of the proposed study. Beyond the number of study lists per session and (therefore) the potential role of interference, the time scales of the retention intervals differ by several orders of magnitude (1.5 minutes vs. 1 day vs. 1 week in the proposed study; 1.5 minutes vs. 12 minutes in the pilot study). Indeed, although the report of the pilot study is helpful in clarifying the general idea and some of the procedural and analytical details (those that remain the same), the relevance of its results as a basis for the current hypotheses is highly questionable. Ironically, this may in fact be fortunate, because had the two studies been more similar, it would then be more problematic to treat hypotheses based on the initial pilot results as representing, in fact, *a priori* hypotheses regarding the expected results of the proposed studies.

More importantly, however, without a clear theoretical rationale for each hypothesis, it is sometimes difficult to evaluate exactly what is being hypothesized and why, and the potential theoretical conclusions that results confirming (or refuting) each hypothesis might yield.

Response: The manuscript has been revised to clarify and address these points (pp. 5-8).

The theoretical rationale for these predictions lies in the previous findings of the all-or-none forgetting framework of contextual information¹¹⁻¹³. This rationale drives the analyses used in the pilot studies. Several different key aspects of the study paradigm are different between pilot study 1 and the proposed study, hence the reasoning for running the proposed study with its suggested paradigm

(namely, a longer delay of one day). The results of pilot study 1 (Appendix A) demonstrate the feasibility of linguistic analysis and that indeed it seems to concur with the hypotheses stemming from the all-or-none forgetting literature. Regarding the feasibility and appropriateness of these findings in a longer time scale, the findings of study pilot 2 (Appendix B) suggest that a similar pattern of results (i.e., preserved justification content) may be found in a longer timescale, and using computational linguistic analyses.

With regard to H3, there is an inconsistency in the precise formulation, between the text appearing on page 5 (of my version of the manuscript) and the one appearing in the summary table (Table 1). In both places it is hypothesized that the number of details per justification will remain stable over time. However, in the text, this prediction is *conditionalized* on the recalled item being correct, whereas in the summary table (and in the formulation of other similar hypotheses) it is not. Without a clear theoretical rationale for this hypothesis, it is difficult to know which version was intended.

Nonetheless, in light of the general rationale alluded to earlier, it appears to be appropriate to restrict all of the stability-related analyses to correct recalls only, in order to include only cases in which the verbalizations reflect the content of actual (and still accessible) memory representations. However, if a substantial number of verbalized justifications are provided for incorrect recalls, this would suggest that justifications can be confabulated, and if so, they might be confabulated for correctly recalled items as well (i.e., they do not necessarily reflect the content of a “retained” and accessible memory representation). This alone is not necessarily a problem, unless the tendency to confabulate/reconstruct memory justifications for correct recalls (as well as incorrect recalls) increases over time, in which case that might contribute to an “illusional” stability in quantity and quality of memory justifications over time.

Response: The hypotheses in the text (pp. 6 - 7) and table (p. 19-20) have been clarified. With regard to the concern the reviewer raises here, we hypothesize that:

(1) The proportion of incorrect items will not increase for items with justifications over time. This is in contrast to what a confabulating explanation might predict.

This is stated in Hypothesis II (pp. 6/20):

“We predict that justifications will reflect objective memory performance, such that items with justifications will be more likely to be accurate recalls compared to items without Justifications (Hypothesis 2). With regard to effects of time, we further predict that the accuracy of items with justifications will not decrease over time, contrary to what would be predicted if justifications of true memories are prone to increased confabulation over time, in which case the proportion of items with justifications which are correct recalls ought to decrease.”

It may even be that the proportion of incorrect recalls will not only hold steady but decrease. This might be explained by the notion that items with justifications are better preserved over time. This is the case in pilot I, but is a post hoc

explanation for these findings and hence not included in our hypotheses for the study.

(2) Moreover, we hypothesize that the content within justifications of items *correctly recalled*, will remain similar in level of detail over time (Hypothesis III). We predicate this on correct recall, because our hypothesis of unchanged content over time stems from the notion of preserved contextual information which serves to validate and justify a recollection. It is difficult to predict the behavior of justification for false memory (which is inherently a confabulation). Our core idea regards recollection reflective of a true memory, and hence we constrain all our predictions to justification of correct recollection.

H4 predicts that the proportion of recalled items accompanied by justifications will be equivalent at longer and shorter delays. Here there is no mention of conditionalizing on correct recall, but for the same reasons just discussed, it would seem appropriate to do so. Unlike H3 (and H5, H6), however, this hypothesis appears to relate to the presence or absence of recollective-contextual content (in accessible memory representations) after different retention intervals, rather than to the fidelity or quality of this content. Given the authors' statement to the contrary (i.e., that this hypothesis relates to "quality" whereas H3 relates to "quantity"), perhaps they are implicitly assuming that the inability to provide a memory justification reflects a situation in which the "quality" of the contextual information has "degraded" to a point that it can no longer be accessed and/or verbalized. This conceptualization, however, appears to negate the basic distinction between the "accessibility" of information, and the quality (e.g., fidelity) of the information that is accessed, which (as discussed earlier) appears to provide the general rationale for the present research questions and hypotheses.

Response: As mentioned, all hypotheses (III-VI) are predicated on correct recalls. Hypothesis IV is to convey that when classifying participant responses as either having some form of justification (J+) or no form of justification to report (J-, e.g., "I don't know why I remembered"), the prevalence of these different categories will remain similar across time delays. This sense of preserved justifications' content is meant by "qualitatively" different/similar. It is not meant to reflect on the quality of the memory for an item (in the sense of better/poorer memory retrieval). This wording has been rephrased (p. 5/7) to avoid the confusion between terms:

"In the current study we ask whether justifications lose fidelity over time, in line with findings of subjective confidence assessments, or whether they remain intact in their degree of vividness, detail, and their content after a long time-delay."

Indeed, the potential *causal* connection between the amount and quality of represented contextual content on the one hand, and the accessibility of that content, together with the "item" content, on the other, raises a critical issue regarding how one might interpret a confirmation of the hypothesized result. Consider again the general hypothesis, that the accessibility of a memory representation declines over time, whereas the content (both item and contextual) of the representations that are/can be accessed at different

points in time, remains stable. If supported by the findings, they might be taken to imply that “accessibility” and “quality” (e.g., fidelity) are *dissociable* aspects of a memory representation, with one, but not the other, being affected by the passage of time. Alternatively, however, such findings might reflect a situation in which the content of some memory representations degrades over time, whereas, for various reasons, the content of other memory representations does not, with the memory representations that do not degrade (or degrade negligibly) being those that remain accessible. This possibility follows from the commonly accepted notion that successful retrieval depends on achieving a sufficiently good match between the available retrieval cues and the stored contextual information. A confirmatory pattern of results would be consistent with either of these two alternatives, yet they could hardly be more different in their broader theoretical implications. I would stop short of asking for a design that could tease these two possibilities apart[*], but I do think that the issue should be raised in the theoretical rationale that introduces and motivates the study’s research questions and hypotheses (and ultimately, in discussing the results, should they turn out as expected).

[*Logically speaking, a crucial aspect that might distinguish the two possibilities is whether the “content” that is being referred to (and examined) is content that can be used as an effective retrieval cue. As mentioned earlier, the presently proposed instructions appear to direct participants specifically toward content that can/did serve as a retrieval cue, rather than to additional content that may also ultimately constitute part of the recollective experience.]

Response: We thank the reviewer for this important distinction, but agree that teasing apart between these two possibilities is premature. Here we wish to establish these hypothesized results. The manuscript has been revised to include a discussion of this important theoretical distinction (p. 5):

“Conversely, an established effect in the memory literature is that successful memory retrieval hinges on proper *overlap* between available retrieval cues and contextual information retained in memory (“Context Reinstatement”^{14–16}). This latter overlap effect entails an alternate explanation for the hypothesized pattern of results. This alternate explanation is at odds with our account that maintained levels of fidelity, vividness and detail of justifications suggest a dissociation between accessibility to contextual information and the fidelity of what information remains accessible. Thus, according to the alternate explanation, the hypothesized pattern of results might be due to the content of some contextual information being prone to degradation over time, whereas other information remains stable over time. The contextual information which does not degrade would remain accessible and with maintained levels of fidelity. If, as hypothesized, an all-or-none pattern of forgetting is found, future studies will be needed to elucidate between these two theoretical accounts.”

This will also be discussed as an option for a follow up study (provided the hypotheses are confirmed).

[H5-H6] It is also hypothesized that the linguistic content of the memory justifications will remain unchanged over time, in terms of concreteness, tentativeness, and certainty [H5], and in terms of unigram (word?) frequencies [H6]. H5 appears to focus on salient

qualitative aspects that might be expected to change over time, whereas the rationale for H6 is less clear. Again, as discussed above, it appears that these hypotheses should also be conditionalized on correct recall. More substantively, given that the memory “triggers” reported by the participants are being treated as (potential) memory justifications, it would be informative to evaluate not only the participants’ certainty/tentativeness (two poles of a single variable?) regarding the veracity of the triggers themselves (indirectly, via the language used to describe them), but also whether such certainty/tentativeness (as well as other qualitative and quantitative aspects of the reported contextual content) translates into certainty/tentativeness (i.e., confidence) regarding the veracity of the recalled item itself.

In other words, I think that a lot of additional valuable information (both theoretical and “practical”) would be gained by adding some measure of subjective confidence in the correctness of the recalled item. This variable could then be correlated with various measured aspects of the verbal justifications to determine which are, in fact, related to a person’s own subjective justification of the veracity of the “target” memory, and whether these relationships (as well as rated subjective confidence itself; see following comment) change over time. [Confidence ratings should perhaps be elicited after the verbal-justification response/phase, in order to minimize the potential contamination of the verbal justifications themselves.]

Response: These hypotheses are indeed also predicated on correct recalls for the reasons described above (regarding hypothesis III). These are additional measures meant at quantifying through linguistic analysis dis/similarity between justifications following different time delays. Following the reviewer’s useful suggestion we added to the proposed paradigm a numeric confidence scale following each verbal justification (pp. 7/17):

“Following each individual justification, participants will be requested to indicate their level of confidence in their recall on a scale from 1 to 6.”.

This was included in Pilot study 2 (Appendix B, p. B3), and the correlations of numeric confidence scores with linguistic markers calculated: The pairwise correlations between numeric confidence scores and linguistic markers were calculated across justifications, and are shown in Table 1. Numeric Confidence scores were significantly correlated with tentativeness, concreteness and proportion of episodic details, but not with certitude or word count. While certitude and tentativeness appear to be opposite ends of the same confidence variable, they are in fact separate due to the nature of the analysis (with the linguistic inquiry and word count “LIWC”). LIWC counts the proportion of words reflecting specific psychological “scales”, and so may find a greater or smaller degree of using words reflecting tentative language (e.g., if, or, any, something), and likewise for words reflecting certitude (e.g., really, actually, of course, real).

Table 1. Pairwise Correlations of numeric confidence scores and Justifications’ Linguistic markers.

	Numeric Confidence	Tentativeness	Certitude	Concreteness	Word-Count
Tentativeness	-0.27 (<.001)				
Certitude	0.03 (.30)	-0.03 (.24)			
Concreteness	0.13 (<.001)	-0.28 (<.001)	-0.09 (<.001)		
Word-Count	-0.01 (.80)	-0.13 (<.001)	-0.00 (.96)	-0.15 (<.001)	
Episodic-Details	0.07 (.02)	-0.14 (<.001)	-0.09 (<.001)	0.31 (<.001)	0.09 (<.001)

Finally, I think that some of the preceding issues warrant a further consideration and elaboration of H2, which, as presently formulated in the proposal, does not explicitly relate to the issue of stability or change over time (but I believe that it should).

H2 holds that items recalled with justification are more likely to be correct than items recalled without justification. This is a straightforward (“safe bet”) hypothesis that follows directly from the well established positive relationship between recollective experience and/or the recall of contextual details, and the accuracy of the recall (or recognition) response per se. What is less straightforward, however, is the extent to which this positive relationship remains stable over time. That is, is the retrieval of recollective/contextual content (i.e., the “raw materials” for memory justifications) equally diagnostic of recall accuracy after longer compared to shorter retention intervals? What should we expect and why (based on relevant previous findings and theoretical inferences, including those relating to H3-H6)? I believe that in the context of the proposed study (particularly given its initial motivation in terms of the need for, and potential value of, epistemic vigilance), this is an important question/hypothesis that needs to be directly addressed. Also, if subjective confidence ratings are added, hypotheses regarding changes in “justification” content and diagnosticity over time can and should be related to hypotheses (and relevant literature) regarding changes in subjective confidence, and its diagnosticity, over time. When this is done, the authors will face an additional challenge in reconciling their hypotheses with various studies in the metamemory literature indicating that that the relationship between subjective confidence and recall accuracy does in fact weaken over time (e.g., Goldsmith, Koriat, and Pansky, 2005; Shapira and Pansky, 2019).

Response: We thank the reviewer for raising a curious distinction between numeric confidence scales and linguistic justifications. As addressed above,

hypothesis II maintains that this relationship between presence of justifications and accuracy will remain stable over time, as would be predicted by the all-or-none forgetting literature, and the notion of justifications as reflecting objective mnemonic recall (in most cases).

Regarding the content of justifications for correct recall and how it is affected over time (Hypotheses III-VI): a nuanced distinction may be necessary between different linguistic markers proposed. Our main hypothesis regarding the content of justifications (H3) is that justifications shall remain similar in terms of detail of the justification. We maintain that this is the case, as supported by the theoretical framework of the all-or-none forgetting of contextual information and as supported by findings from both pilot studies (using manual and automated counts).

As reflected by the metamemory literature, confidence is affected by the passage of time, even when there is correct recall, in which cases leading to a discrepancy between confidence and accuracy. We propose that similar to maintained accuracy measures, the level of detail in verbal justifications will be unaffected by time delay. However, it remains to be seen whether linguistic markers of confidence as expressed in verbal justifications are more in line with subjective confidence as expressed in direct assessments on numeric scales or with objective accuracy measures (this is possible since linguistic analyses might conceivably convey features of verbal expression which may be implicit). Our preliminary findings from pilot study II seem to suggest that the measures of tentativeness and certitude are not significantly affected by time delays, as opposed to numeric confidence scores. Interestingly, only certain linguistic measures (tentativeness, concreteness) were significantly correlated with numeric confidence scores, while other measures were not.

The manuscript has been revised (pp. 5/7-8) to include this perspective of placing justifications with regard to confidence/accuracy, and the implications for hypothesis V:

“In the current study we ask whether justifications lose fidelity over time, in line with findings of subjective confidence assessments, or whether they remain intact in their degree of vividness, detail, and their content after a long time-delay. A *decrease* in fidelity, vividness and detail over time would be expected due to a gradual decaying of underlying contextual information, whereas *maintained* levels of fidelity, vividness and detail would be in line with a dissociation of accessibility from fidelity. In the latter case, despite reduced accessibility to contextual information, that information which is still accessible will remain unchanged in degree of fidelity (demonstrating an “All-or-None” form of forgetting)...

To examine whether the content of justifications remains similar across short and long delays, reflective of a preservation of underlying contextual cues, we propose assessing justifications’ content using linguistic analysis by employing domain-specific lexicons to examine similarity in relevant topics, and frequencies of single words (“unigrams”) for a measure of similarity in words used within justifications. We will thus use lexicons relevant to justifications’ degree of vividness and confidence, such as language concreteness^{17,18}, and tentative versus certain language^{19,20}. In line with the findings in the pilot study, we hypothesize that measures of language concreteness will remain similar ($BF_{01} > 10$) in short and long delays (Hypothesis 5). The degree of certainty and tentativeness in justifications may either decline over time, in line

with previous findings of reported subjective confidence, or may remain unchanged, reflecting an implicit association with preserved accuracy and justifications over time. We further hypothesize that unigram frequencies^{21,22} will be similar ($BF_{01} > 10$) between long and short time-delays (Hypothesis 6), demonstrating a broad similarity in justifications' content between short and long delays.”

Methodological Comments

On the whole, the study appears to be well designed to examine the six related empirical questions presented as Hypotheses 1 to 6. My one major concern, regarding the formulation of the question used to elicit the “memory justification” responses, was discussed earlier (see comment #1, above). However the authors decide to respond to this issue, it is important that the precise wording of the question be clear, as well as any supplementary instructions or examples (to be) given to the participants.

Another potential problem, also mentioned above, concerns the time delay between the recall test and the memory justification elicitation phase. Participants are essentially being asked to “recall their earlier recall” process for each item, introducing potential 2nd-order memory and/or “narrative” contributions to stability or change in the data purported to reflect the 1st-order memory representations (and processing). While it is clear why the authors prefer not to elicit written justifications in real time following the recall of each item, perhaps they might consider using some type of “think-aloud” protocol that would only minimally disrupt the natural recall process, which could then be recorded and analyzed by methods similar to those proposed for the “offline” written verbalizations. [Perhaps both online and offline methods might be used and compared in separate experiments.]

Response: To address this issue, pilot study 2 was run with a between-subjects variable of justification elicitation order. In one condition participants elicited justifications in a separate phase, and in one condition they provided justifications following each item recalled (Appendix B, pp. B3 / B5). The results of pilot study 2 demonstrate that the justification recall condition did not affect recall in terms of accuracy or reliance on temporal order (pp. B5-B7), suggesting that they are valid for comparison. The justification-eliciting order did not significantly affect the level of detail in justifications, nor justifications' concreteness, tentativeness or certitude, or item's numeric confidence scores (pp. B8-B10), but did affect the length of justifications (pp. B7-B8), with the justifications in the separate condition being longer than the mixed condition. These results suggest that invoking 2nd-order memory representation as in a separate justification condition does not seem to detract from the ability to retrieve and report memory justifications. Our results are in line with previous findings demonstrating that confidence judgements are similar when elicited during interviews or separately afterwards²³.

The design includes a single “short delay” of 1.5 minutes, and two different “long delays”—1 day and 1 week (manipulated between participants). Yet, the hypotheses

refer simply to “short vs. long delays.” Do the authors hypothesize that equivalent stability will be observed after 1 week compared to after 1 day?

Response: While potentially it may be the case that similar stability of representation may hold for long delays of different lengths (as 1 day / 1 week), the findings of pilot study 3 (Appendix B, pp. B13-B14) demonstrate that in the proposed paradigm not enough items are recalled following 1 week delay, and hence the study paradigm has been revised to include a long delay of 1 day only (p. 5-6, 15-16).

Morris Goldsmith

Reviewer #3:

Remarks to the Author:

In this Registered Report the authors ask if written reports (“justifications”) of why specific study items were recalled change as a function of study-retrieval delay. The authors state 6 specific hypotheses to be tested, outline a plan for testing each, and describe how alternative outcomes can be assessed. Briefly, the authors have a target N = 200 participants who will study 2 lists of words, which will then be tested after an immediate period and one of 2 different delayed recalls (1 or 7 days). After recall, participants will provide written justifications for each retrieved word, which will be analyzed in several different ways.

The proposed work would be timely. The memory field continues to experiment with naturalistic approaches to research questions, and this manuscript can inform our understanding of a very “traditional” memory task—free recall of studied words—using a “naturalistic” justification procedure. In contrast, a great deal of prior research has focused on metacognitive judgements using ordinal-scale confidence ratings, feelings of knowing, etc. that do not provide nearly as rich a response from participants. The authors make excellent use of emerging options for data collection and analysis, including Prolific to facilitate data collection and natural language tools such as van Genugten & Schacter’s (2022) automated tool for justification transcript scoring. Table 1 provides a clear description of how each empirical question will be asked/assessed, and the target p-values or Bayes Factor scores are set conservatively. The proposed methods are generally well described and appear to be sufficiently powered. In my comments below, I will identify a few areas that would benefit from clarification. My other concerns center around the framing of the questions more than the proposed methods, and I will unpack these in detail as well.

1) Justifications are divided into 8 categories, with the authors citing Unsworth (2017) for category selection. Can I ask why the authors used this approach rather than using the categories from the Autobiographical Interview, which is already used to count the number of details in each description?

Response: The justifications in pilot study 1 were not in English, and were generally short. Hence the adaptation of the Autobiographical Interview was done only for counting details per justification. In the proposed study (as well as in pilot study 2), justifications--in English--will be submitted only to computational linguistic analysis (e.g., the automated autobiographical score tool) to enable a large sample size which would not be possible with manual scoring methods as used in pilot 1 such as the original autobiographical interview.

2) In cases, where a justification relates to more than one category, is it given a weighting of "1" for each relevant category, or is it divided in some way (e.g., does a word with 2 category justifications contribute 0.5 to each category)?

Response: justification categories described in pilot study 1 were counted in each group separately (with a score of 1 in each). These manual categories however will not be used in the proposed study but will be replaced by automated computational tools to assess the number of details, and the content of justifications.

3) If more than one rater will be assigning justifications to different categories, will different raters be trained using pilot data to ensure consistency? What is the minimum interrater reliability the authors plan to achieve?

Response: We thank the reviewer for raising this concern. Manual assignment to categories in the proposed study (as opposed to pilot study 1) will be merely to groups of an item with-justifications/without-justification (J+/J-; p. 13), where "with" is any form of justification recalled and "without" designates no form of justification (e.g., "I don't know why I recalled this item"). We included in the manuscript (p. 14-15) clarifications about the manual coding procedure, and included our minimum interrater reliability to be achieved - Cohen's Kappa of at least 0.8, interpreted as strong agreement between raters (e.g., McHugh, 2012). "The justifications will be manually coded for analysis into one of two exclusive groups: with-justification (J+) (i.e., an item recalled with a specific description such as: "I recalled the pearls because I made up a story about a pearl necklace"), or without-justification (J-) (i.e., an item recalled without an informative description, such as: "I don't know why, I just remembered it"). The inter-rater reliability will be measured to ensure a strong agreement between manual raters, designated by a Cohen's Kappa statistic of at least 0.8²⁴."

4) As initially described by Tulving (1985), auto-noetic consciousness encompasses personal time extending into the past as well as the future. In other words, it allows for "mental time travel" in the form of memories or episodic future thinking/episodic simulation (e.g., Szpunar et al., 2010). Given that the focus of this work is about the subjective re-experiencing of past events, I might encourage the authors to reduce some of their theoretical reliance on auto-noesis and instead to more clearly invoke the reality monitoring framework proposed by Johnson and others (e.g., Johnson et al., 1988 or Johnson, 2006, which is already cited by the authors).

Response: The manuscript's introduction has been revised to incorporate the reality monitoring framework (pp. 2-3):

“Previous studies of retrieval of true memories (i.e. actual past events), and their distinction from false memories (i.e., events imagined or falsely recalled), demonstrate that both these types of memories share much in common, to the point where it has been suggested they are at times indiscernible from one another^{25–27}. However, in most cases there are noticeable differences between true and false memories--most importantly, true memories are associated with higher degrees of recollection. Recollection in this context refers to retrieval of past information along with vivid contextual information from the experience and often accompanied by high confidence. Previous studies have demonstrated subjective phenomenological differences between retrieval of true and false memories^{28–30}, as indicated by subjective ratings^{26,29,31,32}. Retrieval of true memories is accompanied by details relating to sensory perception and embedding of an event in a spatial and temporal context^{26–28,31,33–36}. The subjective differences between retrieval of true and false memories have further been demonstrated by objective measures of neural activity^{27,31,37}, perceptual priming³¹ and linguistic analysis^{38–41}.”

In addition, the theoretical reliance on auto-noetic consciousness has been downgraded (p. 3), focusing on recollection instead:

“These findings suggest that recollection, the accompaniment of memories with contextual information, is an adaptation meant to address the fundamental problem of distinguishing between veridical and non-veridical memories^{7,42–44}. When we recall an event from memory, we do not merely recall its details, but often reinstate our sense of experiencing the event as well (“*auto-noetic consciousness*”^{44,45}). Information regarding one’s internal context includes among other, our attitudes regarding the event (e.g., enjoyment from seeing a painting)⁴⁶ and our internal monologue (e.g., “I wonder who painted it?”)⁴⁷. The ability to reinstate an experience along with accompanying contextual information (internal or external) may serve as an authentication device that helps us determine that we are remembering correctly⁷.”

5) As a related point, there is evidence that the amount of available contextual information associated with a memory may serve the diagnostic function of separating events that have been personally experienced from those that have been imagined (or, presumably, heard from another source). Support for this hypothesis has typically come from ordinal-scale rating comparisons (e.g., D’Argembeau & van der Linden, 2004; Szpunar & McDermott, 2008; Arnold et al., 2011; see McDermott & Gilmore, 2015 for a review). Provided sufficient ELIs occur, a comparison of the details provided for correctly and incorrectly recalled items may allow for a direct test of predictions of this contextual information model that was not previously possible and would further ground this research in recent theory about how humans decide if a given thought reflects a “real” prior experience.

Response: We thank the reviewer for the useful suggestion. Given the relatively low levels of ELIs in recall, there will likely be highly unbalanced groups between items correctly recalled and ELIs. Nonetheless, given a large enough sample, we may be able to gather enough ELIs for such a comparison. We, therefore, reserve the suggestion for an exploratory analysis to be conducted on the proposed study’s data if there are enough ELIs. The manuscript has been revised (p. 3) to

include references to the theory described by the reviewer, as well as recent studies which have conducted similar comparisons between justifications of true and false memories in recognition studies ^{40,41}:

“Previous studies have demonstrated subjective phenomenological differences between retrieval of true and false memories ²⁸⁻³⁰, as indicated by subjective ratings ^{26,29,31,32}. Retrieval of true memories is accompanied by details relating to sensory perception and embedding of an event in a spatial and temporal context ^{26-28,31,33-36}. The subjective differences between retrieval of true and false memories have further been demonstrated by objective measures of neural activity ^{27,31,37}, perceptual priming ³¹ and linguistic analysis ³⁸⁻⁴¹.”

6) The authors note that a first-person perspective may help with decisions about memory accuracy (lines 37-38), but this is an unnecessary oversimplification of the literature given what I understand to be the goals of the proposed experiment. The perspective taken during recall seems to depend in part on the specifics of a provided cue, such as its age or the type of event it relates to (Rice & Rubin, 2009; 2011) and varies in general across individuals (Berg et al., 2021). The relation between vividness (which one might take as a proxy for reinstating experiences) and visual perspective is non-linear (Berg et al. 2021).

Response: The manuscript has been revised (p. 3) for a more careful and nuanced claim, without staking claims about a first person perspective. Namely, the presence of contextual information may serve to verify the validity of a memory, the content of which can be varied and depending on the specifics of a provided cue. Regardless of the specific trigger-cue, the presence of contextual information of varying forms may serve as a validating marker.

7) On Lines 58-61, the authors state that justifications have never been studied with respect to the passage of time. This may be true for complex verbal reports of the type used here, but it glosses over areas such as eyewitness testimony or metacognition that have considered this basic question using confidence ratings or other “simple” justifications for decades (e.g., Shimamura & Squire, 1988; Sauer et al., 2009).

Response: The manuscript has been revised (p. 4) for a more carefully worded assertion, that despite previous studies using various forms of confidence ratings, verbal justifications as conceptualized in the current study have yet to be examined over time:

“As is true for most of the information stored in memory, it is most likely that the contextual cues on which justifications rely are subject to the perils of time ^{8,13,48}. Previous research has demonstrated the effects of time on rankings of confidence and metacognitive judgements ⁴⁹⁻⁵¹. However, despite the importance of verbal justifications in evaluation of memory’s veracity—including in highly consequential settings ⁵²—no study to date has examined how they are affected by the passage of time.”

References

1. Levine, B., Svoboda, E., Hay, J. F., Winocur, G. & Moscovitch, M. Aging and autobiographical memory: dissociating episodic from semantic retrieval. *Psychol. Aging* **17**, 677–689 (2002).
2. Sekeres, M. J. *et al.* Recovering and preventing loss of detailed memory: differential rates of forgetting for detail types in episodic memory. *Learn. Mem.* **23**, 72–82 (2016).
3. van Genugten, R. D. I. & Schacter, D. L. Automated scoring of the autobiographical interview with natural language processing. *Behav. Res. Methods* **56**, 2243–2259 (2024).
4. Conway, M. A. Episodic memories. *Neuropsychologia* **47**, 2305–2313 (2009).
5. Radvansky, G. A., Doolen, A. C., Pettijohn, K. A. & Ritchey, M. A new look at memory retention and forgetting. *J. Exp. Psychol. Learn. Mem. Cogn.* **48**, 1698–1723 (2022).
6. Ritchey, M., Dolcos, F. & Cabeza, R. Role of amygdala connectivity in the persistence of emotional memories over time: an event-related fMRI investigation. *Cereb. Cortex* **18**, 2494–2504 (2008).
7. Mahr, J. & Csibra, G. Why do we remember? The communicative function of episodic memory. *Behav. Brain Sci.* **41**, 1–93 (2017).
8. Sadeh, T. & Pertzov, Y. Scale-invariant Characteristics of Forgetting: Toward a Unifying Account of Hippocampal Forgetting across Short and Long Timescales. *J. Cogn. Neurosci.* **32**, 386–402 (2020).
9. Brady, T. F., Konkle, T., Alvarez, G. A. & Oliva, A. Real-world objects are not represented as bound units: independent forgetting of different object details from visual memory. *J. Exp. Psychol. Gen.* **142**, 791–808 (2013).
10. Schurgin, M. W., Wixted, J. T. & Brady, T. F. Publisher Correction: Psychophysical scaling reveals a unified theory of visual memory strength. *Nat Hum Behav* **5**, 804 (2021).
11. Joensen, B. H., Gaskell, M. G. & Horner, A. J. United we fall: All-or-none forgetting of complex episodic events. *J. Exp. Psychol. Gen.* **149**, 230–248 (2020).

12. Berens, S. C., Richards, B. A. & Horner, A. J. Dissociating memory accessibility and precision in forgetting. *Nat Hum Behav* **4**, 866–877 (2020).
13. Gamoran, A., Greenwald-Levin, M., Siton, S., Halunga, D. & Sadeh, T. It's about time: Delay-dependent forgetting of item- and contextual-information. *Cognition* **205**, 104437 (2020).
14. Smith, S. M. & Vela, E. Environmental context-dependent memory: a review and meta-analysis. *Psychon. Bull. Rev.* **8**, 203–220 (2001).
15. Hockley, W. E. The effects of environmental context on recognition memory and claims of remembering. *J. Exp. Psychol. Learn. Mem. Cogn.* **34**, 1412–1429 (2008).
16. Manning, J. R., Polyn, S. M., Baltuch, G. H., Litt, B. & Kahana, M. J. Oscillatory patterns in temporal lobe reveal context reinstatement during memory search. *Proc. Natl. Acad. Sci. U. S. A.* **108**, 12893–12897 (2011).
17. Brysbaert, M., Warriner, A. B. & Kuperman, V. Concreteness ratings for 40 thousand generally known English word lemmas. *Behav. Res. Methods* **46**, 904–911 (2014).
18. Muraki, E. J., Abdalla, S., Brysbaert, M. & Pexman, P. M. Concreteness ratings for 62,000 English multiword expressions. *Behav. Res. Methods* **55**, 2522–2531 (2023).
19. Tausczik, Y. R. & Pennebaker, J. W. The Psychological Meaning of Words: LIWC and Computerized Text Analysis Methods. *J. Lang. Soc. Psychol.* **29**, 24–54 (2010).
20. Boyd, R. L., Ashokkumar, A., Seraj, S. & Pennebaker, J. W. The development and psychometric properties of LIWC-22. University of Texas at Austin. Preprint at (2022).
21. Kessler, J. S. Scattertext: a Browser-Based Tool for Visualizing how Corpora Differ. *arXiv [cs.CL]* (2017).
22. Bondi, M. & Scott, M. *Keyness in Texts*. (John Benjamins Publishing, 2010).
23. Spearing, E. R. & Wade, K. A. Providing eyewitness confidence judgments during versus after eyewitness interviews does not affect the confidence–accuracy relationship. *J. Appl. Res. Mem.*

- Cogn.* (2022).
24. McHugh, M. L. Interrater reliability: the kappa statistic. *Biochem. Med.* **22**, 276–282 (2012).
 25. Ost, J., Vrij, A., Costall, A. & Bull, R. Crashing memories and reality monitoring: distinguishing between perceptions, imaginations and 'false memories'? *Appl. Cogn. Psychol.* **16**, 125–134 (2002).
 26. Anastasi, J. S., Rhodes, M. G. & Burns, M. C. Distinguishing between memory illusions and actual memories using phenomenological measurements and explicit warnings. *Am. J. Psychol.* **113**, 1–26 (2000).
 27. Bernstein, D. M. & Loftus, E. F. How to Tell If a Particular Memory Is True or False. *Perspect. Psychol. Sci.* **4**, 370–374 (2009).
 28. Johnson, M. K., Foley, M. A., Suengas, A. G. & Raye, C. L. Phenomenal characteristics of memories for perceived and imagined autobiographical events. *J. Exp. Psychol. Gen.* **117**, 371–376 (1988).
 29. Johnson, M. K., Hashtroudi, S. & Lindsay, D. S. Source monitoring. *Psychol. Bull.* **114**, 3–28 (1993).
 30. Johnson, M. K. & Raye, L. Reality Monitoring. *Psychol. Rev.* (2005) doi:10.1007/978-3-540-29678-2_4947.
 31. Jou, J. & Flores, S. How are false memories distinguishable from true memories in the Deese-Roediger-McDermott paradigm? A review of the findings. *Psychol. Res.* **77**, 671–686 (2013).
 32. Johnson, M. K. Reality monitoring: An experimental phenomenological approach. *J. Exp. Psychol. Gen.* **117**, 390–394 (1988).
 33. Johnson, M. K., Bush, J. G. & Mitchell, K. J. Interpersonal reality monitoring: Judging the sources of other people's memories. *Soc. Cogn.* **16**, 199–224 (1998).
 34. Simons, J. S., Garrison, J. R. & Johnson, M. K. Brain Mechanisms of Reality Monitoring. *Trends Cogn. Sci.* **21**, 462–473 (2017).
 35. Marche, T. A., Brainerd, C. J. & Reyna, V. F. Distinguishing true from false memories in forensic contexts: Can phenomenology tell us what is real? *Appl. Cogn. Psychol.* **24**, 1168–1182 (2010).

36. Norman, K. A. & Schacter, D. L. False recognition in younger and older adults: exploring the characteristics of illusory memories. *Mem. Cognit.* **25**, 838–848 (1997).
37. Schacter, D. L. & Slotnick, S. D. The cognitive neuroscience of memory distortion. *Neuron* **44**, 149–160 (2004).
38. Sap, M., Horvitz, E., Choi, Y., Smith, N. A. & Pennebaker, J. Recollection versus Imagination: Exploring Human Memory and Cognition via Neural Language Models. in *Proceedings of the 58th Annual Meeting of the Association for Computational Linguistics 1970–1978* (Association for Computational Linguistics, Online, 2020).
39. Sap, M. *et al.* Quantifying the narrative flow of imagined versus autobiographical stories. *Proc. Natl. Acad. Sci. U. S. A.* **119**, e2211715119 (2022).
40. Gamoran, A., Lieberman, L., Gilead, M., Dobbins, I. G. & Sadeh, T. Detecting recollection: Human evaluators can successfully assess the veracity of others' memories. *Proc. Natl. Acad. Sci. U. S. A.* **121**, e2310979121 (2024).
41. Dobbins, I. G. & Kantner, J. The language of accurate recognition memory. *Cognition* **192**, 103988 (2019).
42. Schacter, D. L. & Addis, D. R. The cognitive neuroscience of constructive memory: remembering the past and imagining the future. *Philos. Trans. R. Soc. Lond. B Biol. Sci.* **362**, 773–786 (2007).
43. Suddendorf, T., Addis, D. R. & Corballis, M. C. Mental time travel and the shaping of the human mind. *Philos. Trans. R. Soc. Lond. B Biol. Sci.* **364**, 1317–1324 (2009).
44. Tulving, E. Memory and consciousness. *Canadian Psychology / Psychologie canadienne* **26**, 1 (1985).
45. Tulving, E. Episodic memory: from mind to brain. *Annu. Rev. Psychol.* **53**, 1–25 (2002).
46. Wheeler, M. A., Stuss, D. T. & Tulving, E. Toward a theory of episodic memory: the frontal lobes and auto-noetic consciousness. *Psychol. Bull.* **121**, 331–354 (1997).
47. Gilead, M., Liberman, N. & Maril, A. 'I remember thinking ...': Neural activity associated with

- subsequent memory for stimulus-evoked internal mentations. *Soc. Neurosci.* **9**, 387–399 (2014).
48. Wixted, J. T. The psychology and neuroscience of forgetting. *Annu. Rev. Psychol.* **55**, 235–269 (2004).
 49. Shimamura, A. P. & Squire, L. R. Long-term memory in amnesia: cued recall, recognition memory, and confidence ratings. *J. Exp. Psychol. Learn. Mem. Cogn.* **14**, 763–770 (1988).
 50. Sauer, J., Brewer, N., Zweck, T. & Weber, N. The effect of retention interval on the confidence–accuracy relationship for eyewitness identification. *Law Hum. Behav.* (2010).
 51. Lin, W., Strube, M. J. & Roediger, H. L., 3rd. The effects of repeated lineups and delay on eyewitness identification. *Cogn Res Princ Implic* **4**, 16 (2019).
 52. Seale-Carlisle, T. M., Grabman, J. H. & Dodson, C. S. The language of accurate and inaccurate eyewitnesses. *J. Exp. Psychol. Gen.* **151**, 1283–1305 (2022).